


# Impact of peatlands on carbon dioxide (CO₂) emissions from the Rajang River and Estuary, Malaysia

Denise Müller-Dum[1], Thorsten Warneke[1], Tim Rixen[2,3], Moritz Müller[4], Antje Baum[2], Aliki Christodoulou[1], Joanne Oakes[5], Bradley D. Eyre[5], and Justus Notholt[1]

5  [1] Institute of Environmental Physics, University of Bremen, Otto-Hahn-Allee 1, 28359 Bremen, Germany
   [2] Leibniz Center for Tropical Marine Research, Fahrenheitstr. 6, 28359 Bremen, Germany
   [3] Institute of Geology, University of Hamburg, Bundesstr. 55, 20146 Hamburg, Germany
   [4] Swinburne University of Technology, Faculty of Engineering, Computing and Science, Jalan Simpang Tiga, 93350 Kuching, Sarawak, Malaysia
10 [5] Centre for Coastal Biogeochemistry, School of Environment, Science and Engineering, Southern Cross University, Lismore NSW 2480, Australia

*Correspondence to:* Denise Müller-Dum, dmueller@iup.physik.uni-bremen.de





**Abstract.** Tropical peat-draining rivers are known as potentially large sources of carbon dioxide ($CO_2$) to the atmosphere due to high loads of carbon they receive from surrounding soils. However, not many seasonally resolved data are available, limiting our understanding of these systems. We report the first measurements of carbon dioxide partial pressure ($pCO_2$) in the Rajang River and Estuary, the longest river in Malaysia. The Rajang River catchment is characterized by extensive peat deposits found

5   in the delta region, and by human impact such as logging, land use and river damming. $pCO_2$ averaged $2919 \pm 573$ µatm during the wet season and $2732 \pm 443$ µatm during the dry season. This is at the low end of reported values for Southeast Asian peat-draining rivers, but higher than values reported for Southeast Asian rivers that do not flow through peat deposits. However, dissolved inorganic carbon (DIC) and $\delta^{13}$C-DIC data did not suggest that peatlands were an important source of inorganic carbon to the river, with an average DIC concentration of $203.9 \pm 59.6$ µmol $L^{-1}$ and an average $\delta^{13}$C-DIC of $-8.06 \pm 1.90$ ‰.

10  Also, compared to rivers with similar peat coverage, the $pCO_2$ in the Rajang was rather low. Thus, we suggest that peat coverage is, by itself, insufficient as sole predictor of $CO_2$ emissions from peat-draining rivers, and that other factors, like the spatial distribution of peat in the catchment and pH, need to be considered as well. In the Rajang River, peatlands probably do not contribute much to the $CO_2$ flux due to the proximity of the peatlands to the coast. $CO_2$ fluxes to the atmosphere were $2.28 \pm 0.52$ gC $m^{-2}$ $d^{-1}$ (wet season) and $2.45 \pm 0.45$ gC $m^{-2}$ $d^{-1}$ (dry season), making the Rajang River a moderate source of carbon

15  to the atmosphere.



## 1 Introduction

Tropical rivers are hotspots of carbon fluxes to the ocean (Dai et al., 2012) and the atmosphere (Aufdenkampe et al., 2011; Raymond et al., 2013). It has been estimated that 78% of riverine carbon dioxide ($CO_2$) emissions occur in the tropics (Lauerwald et al., 2015). Tropical wetlands exert a particularly strong influence on the carbon budget of these rivers. Two

regional studies independently showed that the partial pressure of $CO_2$ ($pCO_2$) in rivers increases with increasing wetland coverage in the catchment: Borges et al. (2015) established this relationship for African rivers and Wit et al. (2015) for Southeast Asian rivers, many of which flow through peatlands. These peatlands represent a unique type of wetland of global importance. The permanently wet, anoxic soil allows for the accumulation of organic matter at rates which make them the most effective terrestrial carbon store on a millennial timescale (Dommain et al., 2011). Southeast Asian peatlands store 68.5

Gt carbon (Page et al., 2011).

Rivers flowing through these peatlands have the highest riverine dissolved organic carbon (DOC) concentrations worldwide (Alkhatib et al. 2007; Moore et al., 2011; Müller et al., 2015), with an annual average of 68 mg $L^{-1}$ DOC found in an undisturbed peat-draining river (Moore et al., 2013). Because of these high DOC concentrations, Indonesian rivers may account for 75 % of the DOC flux into the South China Sea (SCS) (Huang et al., 2017). Surprisingly, $CO_2$ emissions from these rivers are not

exceptionally high (Müller et al., 2015; Wit et al., 2015). This is attributed to a short residence time of the organic matter in the river, allowing little time for decomposition, and the resistance of peat-derived carbon to bacterial degradation.

However, most Southeast Asian peat-draining rivers are disturbed by human activities such as river damming, urbanization, deforestation (Milliman and Farnsworth, 2011) and discharge of untreated wastewater (Park et al., 2018). Anthropogenic change poses a new challenge to understanding carbon fluxes in Asian river systems, and more data are urgently needed to

constrain the carbon budget for this important region (Park et al., 2018). In Malaysia, a country holding the second largest share of tropical peat (Page et al., 2011), river $CO_2$ emissions have only been studied in a small undisturbed peat-draining river (Müller et al., 2015), in estuaries (Chen et al., 2013; Müller et al., 2016) and in two river reaches which were not influenced by peat (Müller et al., 2016). In this study, the longest Malaysian river, the Rajang River on the island of Borneo, was investigated. This river flows through largely logged-over tropical rainforest (Gaveau et al., 2014), urban areas and disturbed

peat swamps (Gaveau et al., 2016). The aim of this study was to assess the Rajang River and Estuary carbon load and to investigate the impact of peatlands on its $CO_2$ emissions. We expected to see a clear peat signal, i.e. elevated $CO_2$ concentrations in the peat area.



## 2 Materials and Methods

### 2.1 Study area

The Rajang River is located in the Malaysian state of Sarawak in the northern part of the island of Borneo (Fig. 1a). Sarawak has a tropical climate with high temperatures (average 26.6°C, 1992-2016 in Sibu, DWD, 2018) and high precipitation (average

3,578 mm yr$^{-1}$, 1992-2016 in Sibu, DWD, 2018). The region experiences two monsoonal periods: the northeastern monsoon with enhanced rainfall and frequent floods occurs between December and February ("wet season", see Fig. 2a), while the southwestern monsoon from May until September is associated with relatively drier weather ("dry season"). However, despite the monsoon seasons, rainfall is high throughout the year (Sa'adi et al., 2017).

The Rajang River originates in the Iran mountains, a mountain range at the border between Malaysia and Indonesia

(MacKinnon, 1996) with elevations of up to 1,800 m (Milliman and Farnsworth, 2011). It drains an area of approximately 51,500 km² (Lehner et al., 2006; DID 2017) whose geology is dominated by Cenozoic sedimentary and metamorphic rocks, consisting of siliciclastic rock with minor amounts of carbonates (Staub et al., 2000; Milliman and Farnsworth, 2011). The Rajang River flows approximately 530 km from east to west and discharges into the South China Sea (Milliman and Farnsworth, 2011). Main settlements along the river are the towns of Kapit, Kanowit and the city of Sibu (163,000 inhabitants)

(see Fig. 1b). In addition, a large number of longhouses (traditional buildings inhabited by local communities) are located along the river and its tributaries (Ling et al., 2017). Hydroelectric power plants were built on two tributaries in the upper Rajang basin: The Bakun hydroelectric power plant commenced operation in 2011 and the Murum dam in 2015 (Sarawak Energy, 2013, see Fig. 1b). The construction of another hydroelectric power plant on a tributary in the southern Rajang basin is planned for the future (Sarawak Energy, 2013).

The Rajang delta system is comprehensively described in Staub and Gastaldo (2003). It is entirely surrounded by peatlands (Fig. 1b), which extend over an area that corresponds to approximately 11% of the catchment size (Nachtergaele et al., 2009). Most of these peatlands have been converted to industrial oil palm plantations (Gaveau et al., 2016, Fig. 1b). The main distributary channels forming the delta (from north to south) are the Igan, Hulu Seredeng (which splits up into Lassa and Paloh), Belawai and Rajang, which have a maximum tidal range (spring tide) of 3-6 m (Staub and Gastaldo, 2003). Saltwater

intrudes into the estuary approximately as far as the point where the Rajang River splits up into its four southernmost distributaries, a few kilometers downstream of Sibu (Fig. 1b), depending on season. Tidal influence extends further inland approximately up to the town of Kanowit (Staub and Gastaldo, 2003).

Monthly discharge (Fig. 2a) was estimated from monthly precipitation (1992-2016; DWD, 2018) and an evapotranspiration rate of 1,545 mm yr$^{-1}$ (Kumagai et al., 2005) or 43.2%. Annual average discharge from 1992-2016 was 3,322 m³ s$^{-1}$ in

reasonable agreement with reported discharges of 3,490 m³ s$^{-1}$ (Milliman and Farnsworth, 2011) and 3,372 m³ s$^{-1}$ for the years 1991-2015 (Sa'adi et al, 2017).





## 2.2 Surveys

We sampled the Rajang River during two surveys, which were designed to get spatial coverage of both peat and non-peat areas during the wettest and driest period of one year. The first survey took place at the peak of the monsoon season in January 2016 ("wet season"). The second one was performed during the dry season in August 2016 ("dry season"). In January 2016, we

entered the Rajang River through the Rajang river mouth (distributary 5 in Fig.1b), went upstream to the town of Kapit and back downstream to the town of Belawai at the Belawai river mouth (distributary 4 in Fig. 1b). In August 2016, we entered the Rajang River through the Rajang river mouth (5), went upstream to Kapit and back to Sibu. From there, we went out to the coast through the Lassa distributary (2), and back to Sibu through the Igan distributary (1). The last sampling stretch was from Sibu into the Paloh distributary (3) and back to Belawai (4). During this campaign, one stationary measurement was performed

overnight in Sarikei in the Rajang distributary in order to assess tidal/diurnal variability.

## 2.3 CO₂ measurements

The setup on the boat was similar to the one described in Müller et al. (2016). Surface water was pumped through a shower-type equilibrator (Johnson, 1999) at a rate of approximately 15 L min$^{-1}$. In the beginning, the equilibrator headspace was connected to an FTIR analyzer (Griffith et al., 2012), which allows for the simultaneous measurement of $CO_2$, methane ($CH_4$),

nitrous oxide ($N_2O$) and carbon monoxide (CO). During the cruise in January 2016, a failure of the FTIR analyzer occurred and measurements were continued (also in August 2016) using an Li-820 non-dispersive infrared (NDIR) analyzer for the measurement of $CO_2$ (Licor, USA). For calibration and inter-calibration of the two instruments, a set of gravimetrically prepared gas mixtures (Deuste Steininger) was measured, which were calibrated against the World Meteorological Organization (WMO) standard scale by the Max Planck Institute for Biogeochemistry in Jena, Germany. For the FTIR, spectra

were averaged over 5 minutes and dry air mole fractions were retrieved using the software MALT5 (Griffith 1996). Li-820 data were stored with a temporal resolution of 1 minute. Gas partial pressure was determined using measurements of ambient pressure with a PTB110 barometer (Vaisala, Finland) and correction for removal of water according to Dickson et al. (2007). Water temperature was measured in the equilibrator and in the surface water and correction to water surface temperature was performed according to Dickson et al. (2007).

In August 2016, the internal pressure sensor of the Li-820 failed. Because the instrument performs an internal correction based on the cell pressure, this correction had to be reversed and recalculated with an assumed internal cell pressure. This procedure is described in the Supplement.

$CO_2$ fluxes ($FCO_2$, in gC m$^{-2}$ d$^{-1}$) across the water-air interface were computed using the gas transfer equation

$$FCO_2 = kK_0(pCO_2^{water} - pCO_2^{air}) \cdot f_1 f_2 \qquad (1),$$

where $k$ is the gas transfer velocity (m s$^{-1}$), $K_0$ is the solubility (mol L$^{-1}$ atm$^{-1}$) calculated according to Weiss (1974), $pCO_2^{water}$ is the partial pressure of $CO_2$ in water, $pCO_2^{air}$ is the partial pressure of $CO_2$ in the overlying air (both in µatm), $f_1$ is a conversion



factor from L$^{-1}$ to m$^{-3}$, and $f_2$ is a conversion factor from µmol s$^{-1}$ to mg d$^{-1}$. The atmospheric mole fractions of CO$_2$ during the months of our measurements were derived from the NOAA ESRL Carbon Cycle Cooperative Global Air Sampling Network (Dlugokencky et al., 2018) for the closest station, which was Bukit Kototabang, Indonesia.

As no information on the tidal currents was available for the Rajang River, we chose the $k$-parameterization by Borges et al.
(2004) ('B04'), which was established for estuaries and considers water flow velocity and wind speed as the main drivers of turbulence, while tidal currents are neglected:

$$k_{600,B04} = 1.0 + 1.719 w^{0.5} h^{-0.5} + 2.58\, u_{10} \tag{2},$$

where $w$ is the water flow velocity (cm s$^{-1}$), $h$ is the depth (m) and $u_{10}$ is the wind speed at 10 m (m s$^{-1}$). Water flow velocity in the lower river reaches was measured by Staub and Esterle (1993) to be 0.7 m s$^{-1}$. A more recent study by Ling et al. (2017)
reports a flow velocity of 1.1 m s$^{-1}$ for the Rajang River upstream from Kapit. For the calculation of the gas exchange velocity $k$, we used the average of 0.9 m s$^{-1}$. Wind speed at 10 m was taken from NOAA NCEP Reanalysis for the grid centered at 2.85°N, 112.5°E for the time of our measurements.

As the gas exchange velocity is critical for the calculation of fluxes, we compared the B04-model to $k$-parametizations by Alin et al. (2011) ('A11') and Raymond and Cole (2001) ('R01'), which were developed for large rivers and estuaries, respectively,
and consider only wind speed as the driver of turbulence:

$$k_{600,A11} = 4.46 + 7.11 \cdot u_{10} \tag{3}$$

$$k_{600,R01} = 1.91 \cdot e^{0.35 u_{10}} \tag{4}$$

**2.4 Ancillary measurements**

In January 2016, individual water samples were taken at 15 stations between the river mouth and Kapit, including the
distributary channels Rajang and Belawai. In August 2016, water samples were taken at 34 stations, with a higher sampling frequency and coverage in the delta (Rajang, Igan, Lassa, Paloh and Belawai, Fig. 1b). Water samples were taken from approximately 1 m below the surface using a Van Dorn water sampler. Particulate material was sampled on pre-weighed and pre-combusted glass fiber filters. From the net sample weight and the volume of filtered water, the amount of suspended particulate matter (SPM) was determined. For POC, 1N hydrochloric acid was added in order to remove inorganic carbon from
the sample. For the determination of carbon, samples were catalytically combusted at 1050°C and combustion products were measured by thermal conductivity using a Euro EA3000 Elemental Analyzer. Repeatability for C content was 0.04 % (standard deviation).

In August 2016, water samples were also taken for the determination of dissolved inorganic carbon (DIC) and the isotopic composition (δ$^{13}$C) of DIC. Samples were poisoned with 200 µL concentrated HgCl$_2$ and filtered through Whatman glass fiber
filters (GF/F, pore size 0.7 µm). 40 ml sampling vials were filled to the top, leaving no headspace, checked for the existence





of bubbles, and stored refrigerated until analysis. Concentrations and $\delta^{13}$C of DIC were measured via continuous flow wet-oxidation isotope ratio mass spectrometry (CF-WO-IRMS) using an Aurora 1030W TOC analyzer coupled to a Thermo Delta V Plus IRMS (Oakes et al., 2010). Sodium bicarbonate (DIC) of known isotope composition dissolved in helium-purged milli-Q was used for drift correction and to verify concentrations and $\delta^{13}$C values. Reproducibility for DIC was ±10 μmol L$^{-1}$ for

concentrations and ± 0.10‰ for $\delta^{13}$C (standard deviations).

During both surveys, dissolved oxygen and water temperature were continuously measured with a temporal resolution of 5 minutes using an FDO 925 oxygen sensor and a WTW 3430 data logger (Xylem Inc., USA). The oxygen sensor was calibrated by the manufacturer, a routine function check was performed before the start of measurements using the check and calibration vessel (FDO © Check) provided by the company. The reported accuracy of a dissolved oxygen measurement at 20°C in air-

saturated water is 1.5%, the precision of the accompanying temperature measurement is 0.2°C (WTW, 2012). pH, salinity and temperature were measured at the stations, using a SenTix 940 pH sensor (pH) and a Multiprobe (Aquaread AP-2000). The pH sensor was calibrated before the start of the measurements using NIST (National Institute of Standards and Technology) traceable buffers. Since salinity was only measured at the stations, we spatially interpolated salinity for the interpretation of $p$CO$_2$ data. This procedure is described in the Supplement. Salinity values ≤2 were considered as freshwater, while we define

estuary as brackish river reaches with salinity >2 but <33. In the following, results are reported for freshwater and estuary separately. Further distinction was made between peat (longitude<112.1°) and non-peat (longitude≥112.1°). This distinction is equivalent to the distinction between tidal river (=peat) and non-tidal river (=non-peat) (Fig. 1b). The terminology used in this study is: peat ($S$≤2, longitude <112.1°), non-peat ($S$≤2, longitude >112.1°), estuary (33>$S$>2). For certain purposes, we report freshwater (peat + non-peat) or delta (peat + estuary) emissions.

**2.5 Data analysis and export calculations**

Data analysis was performed with Python 2.7.15 and ArcMap 10.5. Seasonal differences were tested for significance using the Mann-Whitney U-test from the Python Scipy Statistical Functions module.

In order to calculate the total carbon export from the Rajang River, we derived DOC export, POC export, DIC export and CO$_2$ outgassing for the months of our measurements as follows:

The river load of DOC, POC and DIC was calculated using

$$RIVER\ LOAD = C \cdot Q \cdot f_3 \tag{5},$$

where $C$ is the average concentration of DOC/POC/DIC in mg L$^{-1}$, $Q$ is monthly discharge (m$^3$ s$^{-1}$) and $f_3$ is a conversion factor from s$^{-1}$ to month$^{-1}$.

For DOC, we used the average freshwater DOC concentration reported by Martin et al. (2018) of 2.0 mg L$^{-1}$ (wet) and 2.1 mg L$^{-1}$

(dry). For POC, we used the average freshwater concentrations determined during our surveys.



For the wet season survey, DIC was calculated from pH and $p$CO$_2$ using the program CO$_2$sys (Lewis and Wallace, 1998). Note that pH measurements were only available at the stations, and sometimes we did not have parallel $p$CO$_2$ measurements. Therefore, the number of calculated freshwater DIC values is 9. All errors were calculated with error propagation.

The freshwater CO$_2$ emissions were calculated from the average CO$_2$ flux and the assumption that the river surface area corresponds to 0.89% of the catchment size (average for COSCAT 1328, Raymond et al., 2013). As the river widens substantially in the delta (estuary and peat area), the water surface area in the delta was derived using ArcMap 10.5. The procedure was similar to the one employed by Müller et al. (2016) and is described in the Supplement.

The contribution of non-peat river CO$_2$ to delta emissions was then calculated according to Rosentreter et al. (2018):

$$Nonpeat\ contribution(\%) = (\frac{F_{Nonpeat}}{F_{Delta}} \cdot 100) \tag{6},$$

where $F_{Nonpeat}$ is the lateral CO$_2$ flux from the non-peat area (g d$^{-1}$) and $F_{Delta}$ are the CO$_2$ emissions from the delta (g d$^{-1}$). The lateral CO$_2$ flux from the non-peat area was calculated from riverine excess CO$_2$:

$$Riverine\ excess\ CO_2 = DIC_{Insitu} - DIC_{Equilibrium} \tag{7},$$

where $DIC_{Insitu}$ was the freshwater average from calculated DIC (wet) and from our measurements (dry) and $DIC_{Equilibrium}$ was calculated using CO$_2$Sys.

A non-peat contribution of 100% means that all the emissions in the delta can be explained by ventilation of non-peat CO$_2$. A non-peat contribution of >100% implies that some of the non-peat CO$_2$ is even exported to the ocean, while a non-peat contribution of <100% implies that in addition to non-peat CO$_2$, there are CO$_2$ sources in the delta.

## 3 Results

### 3.1 General characterization of the Rajang River

Measured salinity ranged between 0 and 18.6 during the wet season and 0 and 32.1 during the dry season. Saltwater was detected further upstream during the dry season than during the wet season (Fig. 3a and b). Saltwater penetrated further inland in the Rajang and Belawai distributaries than in the Igan distributary (Fig. 3b), suggesting that most freshwater is discharged via the Igan distributary.

The Rajang River was slightly acidic (average pH = 6.7 and 6.8, see Table 1) and highly turbid, with SPM concentrations of 179 mg L$^{-1}$ (wet season) and 48 mg L$^{-1}$ (dry season) on average (Table 1). With higher SPM during the wet season (p=0.005), the organic carbon content of SPM was significantly decreased (1.6% on average, p=0.0007) if compared to the dry season (2.3 % on average, see Table 1). POC ranged from 0.7 mg L$^{-1}$ to 9.1 mg L$^{-1}$ during wet season (freshwater average 2.9 mg L$^{-1}$, see Table 1) and from 0.3 mg L$^{-1}$ to 1.9 mg L$^{-1}$ during dry season (freshwater average 1.1 mg L$^{-1}$, see Table 1). The seasonal difference was significant (p=0.012).





The river water was consistently undersaturated with oxygen with respect to the atmosphere. DO oversaturation was not observed. DO averaged 76.8 ± 9.9 % (wet season) and 75.0 ± 7.0 % (dry season, see Table 1) and was on average lower in the peat area (73.0% (wet) and 68.1% (dry)) and in the estuary (68.9% (wet) and 74.3% (dry)) than in the non-peat area (81.1% (wet) and 79.8% mg L$^{-1}$ (dry), see Table 2).

Measured DIC in the dry season ranged from 153.7 µmol L$^{-1}$ in the non-peat area to 2399.2 µmol L$^{-1}$ in the estuary and varied linearly with salinity (Fig. 4a). The freshwater average was 203.9 µmol L$^{-1}$ (Table 1). DIC was slightly higher (p=0.044) in the peat area (235.1 ± 74.3 µmol L$^{-1}$) than in the non-peat area (177.9 ± 20.4 µmol L$^{-1}$) and highest in the estuary (1531.1 ± 593.1 µmol L$^{-1}$, Table 2). Calculated DIC for the wet season averaged 301.3 µmol L$^{-1}$. δ$^{13}$C-DIC ranged between -11.87 ‰ and -1.4 ‰ and averaged -8.06 ‰ (freshwater average, Table 1). δ$^{13}$C-DIC was positively correlated with DIC for the estuary

(r = 0.70) and negatively correlated with DIC for the freshwater part (peat and non-peat combined, r = -0.87, Fig. 4b). While Figure 4 indicates that there might be a difference between peat and non-peat samples, the difference was not significant due to the lack of samples.

## 3.2 Carbon dioxide

The Rajang River was found to be oversaturated with CO$_2$ with respect to the atmosphere, with an average freshwater $p$CO$_2$

of 2919 µatm (wet season) and 2732 µatm (dry season, see Table 1). The $p$CO$_2$ and its spatial distribution were strikingly similar during the wet and dry seasons (Fig. 3c and d). $p$CO$_2$ was significantly higher (p<0.0001) in the peat area (3472 µatm (wet) and 3053 µatm (dry)) than in the non-peat area (2531 µatm (wet) and 2337 µatm (dry), see Table 2 and Fig. 3c and d). In the estuary, $p$CO$_2$ was lower during the wet season (2046 µatm) with an average estimated salinity of 16.5, and higher during the dry season (2608 µatm) with an average estimated salinity of 25.0. Note that this difference may reflect the different

sampling strategies (more and different distributaries were included in the dry season survey). Tidal variability of $p$CO$_2$ was observed at an overnight station in Sarikei in August 2016. During this time, $p$CO$_2$ increased from approximately 3000 µatm to almost 6000 µatm during rising tide (not shown), so the timing of our measurements in the delta relative to the tidal conditions probably also impacted the average values for the estuary.

$p$CO$_2$ decreased with increasing salinity in the estuary (Fig.3). However, a big spread of data in both the high-salinity region

(during the tidal measurement described above) and the freshwater region was observed. $p$CO$_2$ was correlated with DO (Fig. 5). An interesting pattern is consistently visible in both the wet and dry season data, by which main stem data can clearly be distinguished from those collected in the Belawai and Paloh distributaries. Night time measurements beyond the tidal part of the river were too few to make a sound statement about a difference between day- and night time $p$CO$_2$ and DO.

Wind speed in the grid centered at 2.85°N, 112.5°E averaged 0.57 m s$^{-1}$ during our campaign in January 2016 and 1.09 m s$^{-1}$

during our campaign in August 2016 (Table 2). The calculated gas exchange velocities for a Schmidt number of 600 ($k_{600, B04}$) were 8.23 cm h$^{-1}$ and 9.57 cm h$^{-1}$, respectively. This compares to the A11 model with 8.51 cm h$^{-1}$ and 12.19 cm h$^{-1}$ and to the



R01 model with 2.32 cm h$^{-1}$ and 2.79 cm h$^{-1}$ for the wet and dry season, respectively. Fluxes reported in this study are calculated from the B04 model, which yielded intermediate values. It was chosen because it recognizes flow velocity as a driver of turbulence in addition to wind speed. Results for the other two models are compared in the Supplement. The resultant $CO_2$ fluxes ($FCO_2$) to the atmosphere were 2.28 ± 0.52 gC m$^{-2}$ d$^{-1}$ in the wet season and 2.45 ± 0.45 gC m$^{-2}$ d$^{-1}$ in the dry season

(per water surface unit area, see Table 2).

### 3.3 Carbon river load and CO₂ emissions

Discharge was above average in the years 2016 and 2017 (Fig. 2a). Discharge during January 2016 (wet season) was in accordance with the long-term average, but discharge during August 2016 (dry season) was higher than usual. The Rajang River loads were 146 ± 45 GgC in January 2016 and 99 ± 25 GgC in August 2016 (Table 3). Of this, 78 ± 5% (wet) and 65 ±

7% (dry) were exported laterally by discharge. Approximately half of the carbon river load was in the organic form (58 ± 27% and 57 ± 17%, wet/dry). River loads were similar during both seasons, except that POC export was 3-fold higher in the wet season. $CO_2$ emissions to the atmosphere accounted for 22 ± 5% and 35 ± 7% (wet/dry) of the total carbon load of the river and 55 ± 24% and 59 ± 24% (wet/dry) of the combined $CO_2$+DOC export ($CO_2$/DOC flux ratio as calculated in Wit et al., 2015). The non-peat contribution to delta emissions was 126 ± 66 % in the wet season and 54 ± 52 % in the dry season.

## 4. Discussion

### 4.1 Organic carbon load and sediment yield

The proportion of laterally transported carbon in the Rajang River that is in organic form (58 ± 27% and 57 ± 17%, wet/dry) is similar to what has been reported for the carbon flux to the South China Sea (50 ± 14%, Huang et al., 2017). Likewise, the $CO_2$/DOC flux ratio of 55 ± 24% and 59 ± 24% (wet/dry) is in agreement with the average for Southeast Asian rivers of 54 ±

7% (Wit et al., 2015).

The Asia-Pacific region is known for its high sediment yields, especially where rivers drain Cenozoic sedimentary and volcanic rock (Milliman and Farnsworth, 2011). Therefore, the high suspended sediment load in the Rajang River is not surprising. However, SPM concentrations during our expeditions were substantially lower (179.2 mg L$^{-1}$ and 47.8 mg L$^{-1}$) than in July 1992 (613 mg L$^{-1}$, Staub and Esterle, 1993). This could be an effect of upstream dams (operational since 2011 and 2015),

which trap sediment in their reservoirs, thereby reducing downstream sediment loads (Vörösmarty et al., 2003, Snoussi et al., 2002). In support of this, SPM concentrations were intermediate in the upper Rajang River in 2014/2015 (218.3 mg L$^{-1}$, Ling et al., 2017). These measurements were taken before, and the measurements in the current study were taken after, the Murum dam began full operation in the second quarter of 2015. Furthermore, SPM and POC concentrations (2.9 mg L$^{-1}$ and 1.1 mg L$^{-1}$) in the Rajang River were similar to those in the Pearl River, China, (SPM: 70 mg L$^{-1}$ -247 mg L$^{-1}$, POC: 1.0 mg L$^{-1}$ -3.8 mg




$L^{-1}$, Ni et al., 2008) and the Red River, Vietnam, (SPM: 294 ± 569 mg $L^{-1}$ (wet) and 113 ± 428 mg $L^{-1}$ (dry), POC: 3.7 ± 2.0 mg $L^{-1}$ (wet) and 1.1 ± 1.1 mg $L^{-1}$ (dry), Le et al., 2017), both of which are also affected by damming.

SPM was higher during the wet season than during the dry season in agreement with observations at the Kinabatangan River, Malaysia (Harun et al., 2014). This can be attributed to enhanced erosion during the wet season. In logged-over forest, as found in most of the Rajang River basin, the energy impact of rain drops on the soil is higher than in densely vegetated areas, where rain drops are intercepted by the canopy before falling on the ground (Ling Lee et al., 2004). In agreement with this line of reasoning, Ling et al. (2016) showed that the amount of suspended solids in Malaysian streams draining areas with logging activities increased significantly after rain events. The decreased organic carbon content observed during the wet season further supports this, as it indicates a higher contribution of eroded mineral soil to SPM. This pattern is observed in many rivers in this region (Huang et al., 2017). Despite the changing carbon content, most POC was still exported during the wet season, as in other Southeast Asian rivers (Ni et al., 2008; Moore et al., 2011).

### 4.2 Inorganic carbon load

### 4.2.1 DIC concentrations and sources

DIC concentrations in the Rajang River (203.9 µmol $L^{-1}$ in the dry season and 301.1 µmol $L^{-1}$ in the wet season) were comparable to those reported by Huang et al. (2017) for the Rajang River (201 µmol $L^{-1}$ and 487 µmol $L^{-1}$), but substantially lower than reported for the Mekong River (1173-2027 µmol $L^{-1}$, Li et al., 2013) and the Pearl River (1740 µmol $L^{-1}$, Huang et al., 2017). DIC concentrations in the Rajang are similar to those in the Musi River, Indonesia (250 µmol $L^{-1}$, Huang et al., 2017), suggesting that the Rajang River compares better to the equatorial Indonesian rivers than to rivers draining mainland Southeast Asia, probably because of the scarcity of carbonate rock, which has the highest weathering rate and is thus responsible for high DIC in rivers (Huang et al., 2012).

The source of DIC varied along the length of the Rajang River. In the estuarine part, the linear relationship between DIC and salinity (Fig. 4a) suggests that the main source of DIC in the estuary is marine. This is also supported by the relatively high $\delta^{13}$C-DIC of estuarine samples, as ocean DIC is more enriched with $\delta^{13}$C-DIC between 0 and 2.5 ‰ (Rózanski et al., 2003). The positive relationship between DIC and $\delta^{13}$C thus implies an increasing contribution of marine DIC.

In the freshwater part of the Rajang River, $\delta^{13}$C values were more depleted (-8.1 ‰ on average), but higher than those reported for the Lupar and Saribas Rivers in Sarawak (-15.7 ‰ to -11.4 ‰, Müller et al., 2016) or for the Musi, Indragiri and Siak Rivers in Indonesia (-22.5 ‰ to -9.0 ‰, Wit, 2017). Groundwater DIC usually has a $\delta^{13}$C of -16 ‰ to -11 ‰ (Rózanski et al., 2003), depending on the soil $CO_2$ and the weathered rock material. While DIC from silicate weathering is more depleted, DIC from carbonate weathering is more enriched due to the contribution of carbonate-C to DIC (Das et al., 2005). Atmospheric $CO_2$, as present in rain water, has a $\delta^{13}$C of around -8.3 ‰ (at Bukit Kototabang, Indonesia, in 2014, White et al., 2018). In-stream processes affect $\delta^{13}$C-DIC as well: respiration of DOC decreases $\delta^{13}$C, while photosynthesis increases $\delta^{13}$C (Rozanski



et al., 2003; Campeau et al., 2017). $CO_2$ evasion gradually leads to higher $\delta^{13}C$ values until in equilibrium with the atmosphere (with $\delta^{13}C$-DIC around +1 ‰, Polsenaere and Abril, 2012).

It can be assumed that DIC in the Rajang River stems from a combination of these sources. The relatively high $\delta^{13}C$ in our samples suggests that carbonate weathering could play a role; however, only minor amounts of carbonates are present in the

catchment (Staub et al., 2000). The contribution of rain-DIC to total DIC is non-negligible for a river with relatively low DIC (203.9 µmol L$^{-1}$), particularly in a catchment with heavy rainfall that leads to surface runoff, diluting DIC and enhancing $\delta^{13}C$-DIC. With regard to in-stream processes, photosynthesis can be assumed to be negligible in the Rajang River due to its high turbidity, while respiration seems to be important due to the correlation of DO and $pCO_2$ (Fig. 5). This assumption is also supported by the negative correlation of $\delta^{13}C$ and DIC for freshwater samples, because with increasing DIC, $\delta^{13}C$ values get

more depleted, suggesting that organic carbon (with a $\delta^{13}C$ of around -26 ‰ for C3 plants, Rózanski et al., 2003) is respired to $CO_2$ within the river. The overall relatively high $\delta^{13}C$ values can be explained by enrichment due to evasion of $CO_2$. Since we sampled the lower river reaches, it can be assumed that a large fraction of $CO_2$ had already been emitted from the river surface, leading to gradually higher $\delta^{13}C$-DIC.

### 4.2.2 $pCO_2$

$pCO_2$ in the Rajang River (2919 µatm and 2732 µatm (wet/dry)) was higher than in most Southeast Asian rivers without peat influence, such as the Mekong River ($pCO_2$= 1090 µatm, Li et al., 2013), the Red River ($pCO_2$= 1589 µatm, Le et al., 2017) or the freshwater parts of the Lupar and Saribas Rivers ($pCO_2$= 1274 µatm and 1159 µatm, Müller et al., 2016, Table 4, Fig. 6). This could be attributed to the peat influence. However, the Rajang River has a $pCO_2$ at the low end of values reported for peat-draining rivers (Fig. 6): Wit et al. (2015) report values between 2400 µatm in the Batang Hari (peat coverage = 5%) and

8555 µatm in the Siak River (peat coverage = 22%).

A meaningful comparison is also the one between the Rajang River and the Indragiri River, Indonesia, because they have a similar peat coverage (Rajang: 11%, Indragiri: 12%) and peat coverage has previously been considered as a good predictor of river $CO_2$ emissions (Wit et al., 2015). However, $pCO_2$ in the Indragiri (5777 µatm) was significantly higher than in the Rajang, which can be attributed to a lower pH (6.3, numbers from Wit et al., 2015). To illustrate this, we ran a simple exercise using

$CO_2$Sys. At the given temperature, salinity and pH, the $pCO_2$ of 5777 µatm in the Indragiri corresponds to a DIC value of 327 µmol L$^{-1}$. At a hypothetical pH of 6.8, as measured in the Rajang River, this DIC value corresponds, under otherwise unchanged conditions, to a $pCO_2$ of 2814 µatm – which is very close to the average values measured in the Rajang River. This shows that pH is a major determinant for a river's $pCO_2$ (Ruiz-Halpern et al., 2015), and that the peat coverage in a river basin is insufficient as sole predictor of $CO_2$ fluxes. Rather, pH must be taken into account as well, and its drivers must be considered.

$\delta^{13}C$-DIC in the Indragiri was lower (-16.8 ‰, Wit, 2017) than in the Rajang (-8.1 ‰), implying that respiratory $CO_2$ is more dominant in the Indragiri, while the Rajang might be more strongly influenced by weathering, which could explain the higher



pH. Note also that peat coverage is usually reported for the entire catchment (e.g., Wit et al, 2015; Rixen et al., 2016) and does not reveal how much peat is found in estuarine vs. freshwater reaches, which makes comparisons more difficult.

While DOC and $pCO_2$ are positively related to discharge in most rivers (e.g., Bouillon et al., 2012), this pattern is sometimes reversed in peat-draining rivers. This is due to dilution, when rainfall exceeds the infiltration capacity of the wet soil and water runs off at the surface (Clark et al., 2008; Rixen et al., 2016). In the Rajang River, $pCO_2$ was slightly higher during the wet season in agreement with many non-peat-draining tropical rivers (Bouillon et al., 2012; Teodoru et al., 2014; Scofield et al., 2016). However, the seasonality of $pCO_2$ was small, similar to other Malaysian rivers (Müller et al., 2016) and in line with the small seasonal variability of DOC concentrations in the Rajang River (Martin et al., 2018).

Due to an El Nino event, temperatures in Southeast Asia were unusually high in late 2015 and 2016, with a temperature extreme in April 2016 prevailing in most of Southeast Asia (Thirumalai et al., 2017) and unusually hot conditions also recorded in Sibu (Fig. 2b). Given that weathering rates increase at elevated temperatures, DIC from weathering could have been enhanced over other years, although this seems unlikely because DIC was relatively low and in agreement with previous studies. Another factor to be taken into account is that decomposition in the dry upper soil layer is more intense at higher temperatures, and with incipient rainfall, all the resultant DOC is flushed out to the river. Therefore, it is possible that during the year of our measurements, DOC concentrations were higher than usual, and respiratory $CO_2$ may therefore have been enhanced compared to other years.

### 4.2.3 Impact of the peatlands on the $CO_2$ emissions from the Rajang River

The fact that $pCO_2$ was significantly higher in the peat area than in the non-peat area implies, at first glance, that the peat areas are a source of $CO_2$ to the river. However, the difference between peat and non-peat $pCO_2$ has to be interpreted with caution, as the entire peat area is under tidal influence (Fig. 1b). In the following, we will present several arguments that suggest that the peatlands exert only a small influence on the $CO_2$ emissions from the Rajang River.

### a) DOC

One indicator of peatland influence on a river's carbon budget is DOC. DOC concentrations in the Rajang River delta were reported to range between 1.4 mg L$^{-1}$ and 3 mg L$^{-1}$ (Martin et al., 2018). This is at the low end of the range of DOC concentrations reported for peat-draining rivers in Indonesia: These range from 2.9 mg L$^{-1}$ in the Musi River (peat coverage = 3.5%) up to 21.9 mg L$^{-1}$ in the Siak River (peat coverage = 22%; Wit et al., 2015). Rivers whose catchment area is entirely covered by peat exhibit even higher DOC concentrations, with 52 mg L$^{-1}$ (wet season) and 44 mg L$^{-1}$ (dry season) in the Sebangau River, Indonesia (Moore et al., 2011), and 44 mg L$^{-1}$ in the Maludam River, Sarawak, Malaysia (Müller et al., 2015). The Rajang River compares rather to rivers like Lupar and Saribas, Malaysia, which exhibit DOC concentrations of 1.8 mg L$^{-1}$ and 3.7 mg L$^{-1}$ in their freshwater parts (no peat influence, Müller et al., 2016). Consequently, DOC concentrations imply that the peatlands' influence on the Rajang's DOC is rather small.



### b) Non-peat contribution

The non-peat contribution (as calculated according to Eq. (6) and (7)) is a measure of the fraction of delta $CO_2$ emissions that can be explained by upstream (non-peat) sources alone. This means that the non-peat contribution provides an indication of how important $CO_2$ sources within the delta (i.e., peat) are compared to upstream sources. During the wet season, the non-peat

contribution was >100% ($126 \pm 66\%$), suggesting that upstream sources are sufficiently strong to explain all the $CO_2$ emissions in the delta, and that part of the upstream $CO_2$ was even transported to the ocean. However, this does not necessarily mean that there were no additional sources in the delta, as it is unknown how much $CO_2$ was exported to the ocean. During the dry season, the contribution of non-peat $CO_2$ was <100% ($54 \pm 52\%$) suggesting that upstream sources cannot explain all of the $CO_2$ emissions in the delta and that the remainder is derived from within the delta, i.e. net heterotrophy in the peat area and estuary.

Note that the calculated non-peat contributions have relatively large uncertainties, so these statements cannot be made with certainty.

### c) Mixing model

An alternative approach is to theoretically calculate the increase in Rajang River $pCO_2$ that would result from the influx from peatlands. For this, we created a simple model to simulate the mixing of two water masses (see Supplement), one with a pH

of 6.8 and a $pCO_2$ of 2434 µatm (designed to resemble the Rajang River, non-peat area) and the other with a pH of 3.8 and a $pCO_2$ of 8100 µatm (designed to resemble peat-draining tributaries, based on values for the peat-draining Maludam River in Sarawak, Müller et al., 2015). For simplicity, we assumed that mixing occurs at salinity = 0 and that the temperature in both water bodies is the same (28.4°C). From these values, DIC and total alkalinity (TA) of the two water bodies were calculated using $CO_2$Sys. Since they can be assumed to be conservative, we simply calculated DIC and TA of the mixture as

$$DIC_{S=0} = (1 - pc) \cdot DIC_1 + pc \cdot DIC_2 \text{ and } TA_{S=0} = (1 - pc) \cdot TA_1 + pc \cdot TA_2 \qquad \text{(8) and (9),}$$

where $pc$ is the peat coverage in the basin ($pc$=0.11). From DIC and TA, the $pCO_2$ of the mixture was computed ($pCO_2$ = 3058 µatm). This means that if all peat-draining tributaries in the Rajang River basin had a $pCO_2$ of 8100 µatm and a pH of 3.8, the $pCO_2$ in the peat area would be enhanced by around 600 µatm. However, this increase in $pCO_2$ is obviously gradual. For example, at the city of Sibu, peat coverage was estimated at around 2%, for which the described mixing model yields a $pCO_2$

of 2548 µatm (Fig. S3). In the estuary, dilution with sea water already plays a role. Therefore, the mixing model was extended, assuming that at $pc$=3%, salinity is still zero and then linearly increases until $pc$=11%, $S = 32$, $DIC_{S=32} = 2347$ µmol L⁻¹ and $TA_{S=32} = 2324$ µmol L⁻¹ (two end-member mixing model, see Supplement). As a result, $pCO_2$ would theoretically not exceed 2605 µatm if the peat-draining tributaries were the only source of $CO_2$ in the delta (Fig. S3).

However, $pCO_2$ in the peat area was 3472 µatm (wet) and 3053 µatm (dry), so there must be another source of $CO_2$. Since

$pCO_2$ in Sarikei varied 2-fold with the tidal cycle, it seems likely that a large part of the difference in $pCO_2$ between non-peat area and peat area is actually a difference between river and tidal river. Tidal variability is often seen in estuaries (e.g., Bouillon



et al., 2007, Oliveira et al., 2017), largely due to conservative mixing. However, during rising tide in Sarikei, we observed $pCO_2$ values of almost 6000 µatm, which is higher than the freshwater end-member, suggesting that other effects also play a role. Among those are decomposition of organic matter in intertidal sediments (Alongi et al., 1999, Cai et al., 1999) and subsequent transport of the produced $CO_2$ to the river, as well as groundwater input (Rosentreter et al., 2018).

**d) $\delta^{13}$C-DIC**

If the peatlands acted as a significant source of $CO_2$ to the Rajang River, this would also have to be visible in the $\delta^{13}$C-DIC values. In the peat-draining Maludam River, $\delta^{13}$C-DIC averaged -28.55 ‰ (Müller et al., 2015). Thus, it would be expected that the influx of peat-draining tributaries to the Rajang River would decrease $\delta^{13}$C-DIC. Although $\delta^{13}$C-DIC in the Rajang River appeared lower in the peat area than in the non-peat area (Fig. 4b), this difference was not significant. We were therefore

unable to discern a large impact of peatlands on the DIC budget of the Rajang River. It is possible that, because the peatlands are located close to the coast in this system, mixing with sea water occurs before significant effects on the $pCO_2$ are theoretically possible. This means that not only the peat coverage in the catchment is relevant, but also how much of this peat is found in the estuarine reaches. These findings support the arguments of Müller et al. (2015) and Wit et al. (2015) that material derived from coastal peatlands is swiftly transported to the ocean, explaining why peat-draining rivers may not

necessarily be strong sources of $CO_2$ to the atmosphere.

## 5. Conclusions

The Rajang River is a typical Southeast Asian river, transporting large amounts of terrestrial material to the South China Sea. The derived fractions of evaded versus laterally transported carbon are in agreement with other rivers draining into the South China Sea. In contrast to other Southeast Asian rivers with similar peat coverage, the impact of the peatlands on the Rajang

River's $pCO_2$ appeared to be rather small, probably due to the proximity of the peatlands to the coast. As a consequence, $CO_2$ emissions from the Rajang River were moderate compared to other Southeast Asian rivers and low if compared to Southeast Asian peat-draining rivers.

**Data availability**

Calibrated data used in this manuscript are available as a Supplementary Table to this manuscript. Raw data are available at

the Institute of Environmental Physics, University of Bremen, Bremen, Germany.





**Author contribution**

DMD, TW, TR and MM designed this study. DMD performed all measurements and sample preparations during the first survey, AC, AB and MM performed the measurements and sample preparations during the second survey. JO and BE analyzed the DIC and isotopic data, DMD analyzed all other data. All co-authors contributed to the interpretation and discussion of the results. DMD prepared the manuscript with contributions from all co-authors.

**Competing interests**

The authors declare that they have no conflict of interest.

**Acknowledgements**

We would like to thank the Sarawak Biodiversity Centre for permission to conduct research in Sarawak waters (permit no.

10  SBC-RA-0097-MM and export permits SBC-EP-0040-MM and SBC-EP-0043-MM). We also thank Dr. Aazani Mujahid (University of Malaysia Sarawak, Kuching, Malaysia) for her extensive organizational help in preparing the campaigns. Further, we would like to thank all students and scientists from Swinburne University of Technology and UNIMAS in Malaysia for their help during the sampling trips. Lukas Chin and the "SeaWonder" crew are acknowledged for their support during the cruises. The study was supported by the Central Research Development Fund of the University of Bremen, the MOHE FRGS

15  Grant (FRGS/1/2015/WAB08/SWIN/02/1), SKLEC Open Research Fund (SKLEC-KF201610) and ARC Linkage Grant LP150100519.



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





**Figure 1: Map of the Rajang basin on the island of Borneo (a) and a close-up of the basin (b) including the location of the peatlands (Nachtergaele et al., 2009), industrial oil palm plantations (Gaveau et al., 2016) and the stations during the cruise. The distributaries are marked with numbers: 1-Igan, 2-Lassa, 3-Paloh, 4-Belawai, 5-Rajang.**



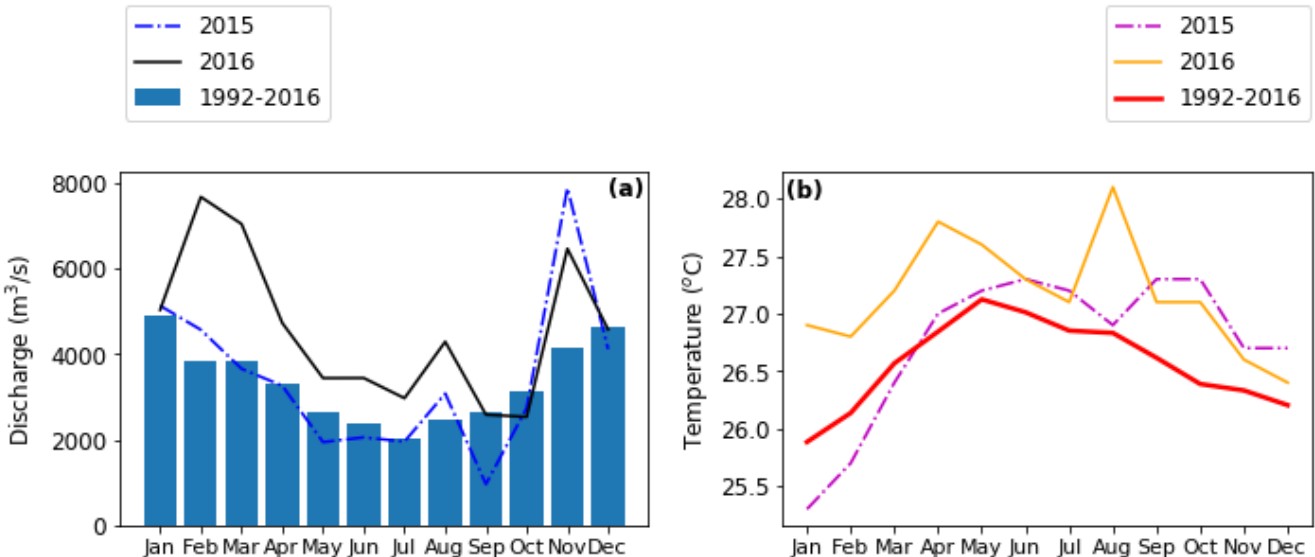

**Figure 2: Monthly discharge calculated for the Rajang River (a) and average temperatures (b) for the city of Sibu (2° 20' N, 111° 50' E) for the years 2015, 2016 and the long-term average from 1992-2016. Data from DWD (2018).**





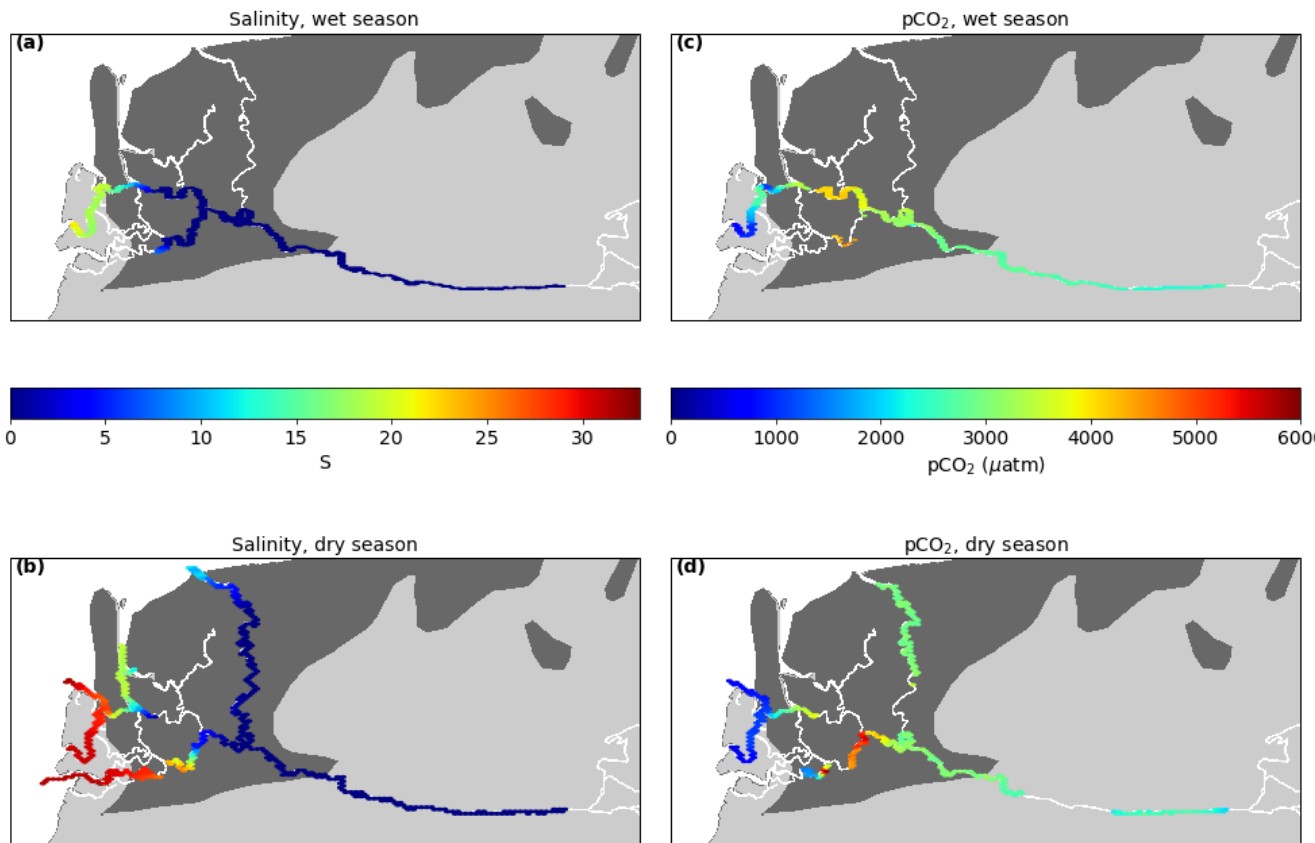

**Figure 3: Salinity (interpolated) and *p*CO₂ distribution in the Rajang River and delta during the wet season survey (a,c) and dry season survey (b,d).**



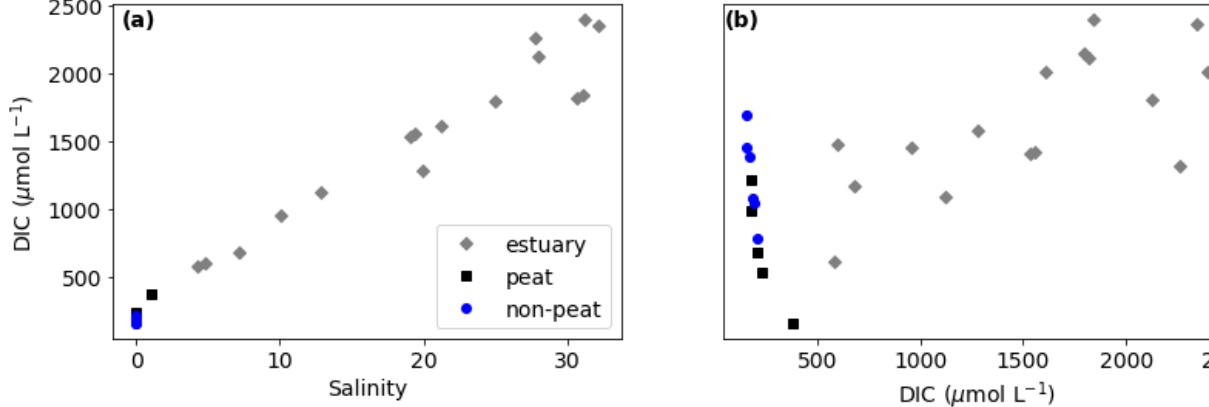

**Figure 4: DIC versus salinity (a) and δ¹³C versus DIC for the estuary and freshwater (=peat + non-peat) samples, respectively. All data refer to dry season samples.**



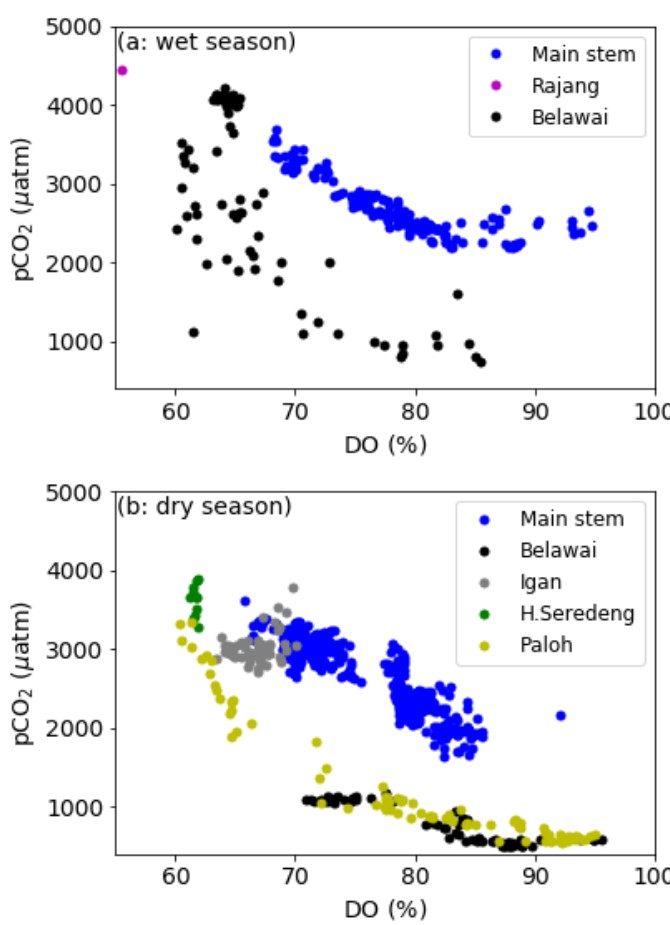

**Figure 5: Correlation of $p$CO₂ versus dissolved oxygen (DO) in the wet and dry season for individual distributaries.**



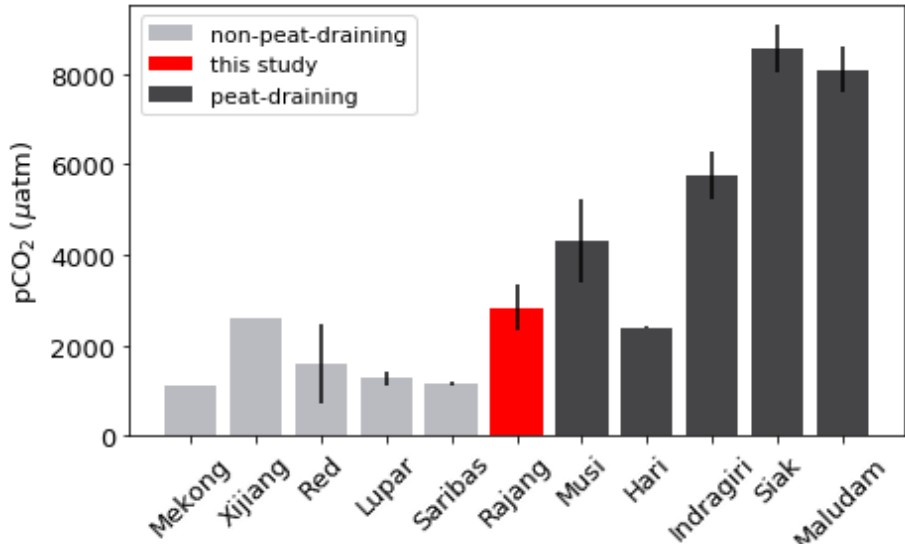

**Figure 6: Comparison of average $p$CO$_2$ values in Southeast Asian rivers. Colors distinguish peat-draining rivers from non-peat-draining rivers. The Rajang River (this study) is highlighted in red.**





**Table 1: Average values for the freshwater part of the Rajang River (salinity ≤2) ± 1 standard error (SE). Results for $k_{600}$ and $FCO_2$ are based on the B04 $k$-parameterization. *denotes a calculated, not measured value.**

|  | **Wet** | **Dry** |
|---|---|---|
| **DO (%)** | 76.8 ± 9.9 | 75.0 ± 7.0 |
| **pH** | 6.7 ± 0.1 | 6.8 ± 0.1 |
| **T (°C)** | 27.5 ± 0.3 | 29.2 ± 0.9 |
| **SPM (mg L$^{-1}$)** | 179.2 ± 74.7 | 47.8 ± 17.1 |
| **POC (mg L$^{-1}$)** | 2.9 ± 1.9 | 1.1 ± 0.4 |
| **%OC in SPM** | 1.6 ± 0.5 | 2.3 ± 0.5 |
| **$p$CO$_2$ (µatm)** | 2919 ± 573 | 2732 ± 443 |
| **DIC (µmol L$^{-1}$)** | 301.3 ± 44.4* | 203.9 ± 59.6 |
| **δ$^{13}$C-DIC (‰)** | n.d. | -8.06 ± 1.90 |
| **u$_{10}$ (m s$^{-1}$)** | 0.57 | 1.09 |
| **k$_{600}$ (cm h$^{-1}$)** | 8.23 | 9.57 |
| **$F$CO$_2$ (gC m$^{-2}$ d$^{-1}$)** | 2.28 ± 0.52 | 2.45 ± 0.45 |





**Table 2: Differences between peat, non-peat and estuary samples (mean ± SE).**

| | $pCO_2$ (µatm) | | $FCO_2$ (gC m$^{-2}$d$^{-1}$) | | $O_2$ (%) | | DIC (µmol L$^{-1}$) |
|---|---|---|---|---|---|---|---|
| | **Wet** | **Dry** | **Wet** | **Dry** | **Wet** | **Dry** | **Dry** |
| **Non-peat** | 2531 ± 188 | 2337 ± 304 | 1.93 ± 0.17 | 2.05 ± 0.32 | 81.1 ± 5.4 | 79.8 ± 3.5 | 177.9 ± 20.4 |
| **Peat** | 3472 ± 477 | 3053 ± 224 | 2.78 ± 0.43 | 2.78 ± 0.22 | 73.0 ± 11.3 | 68.9 ± 7.9 | 235.1 ± 74.3 |
| **Estuary** | 2046 ± 856 | 2607 ± 1763 | 1.38 ± 0.75 | 2.04 ± 1.64 | 68.9 ± 7.9 | 74.3 ± 12.9 | 1531.1 ± 593.1 |



**Table 3: Average discharge and calculated carbon loads and emissions to the atmosphere, estimated for the months of the two surveys and annually.**

|  | **Jan 2016** | **August 2016** | **Annual estimate** |
|---|---|---|---|
| **Discharge (m³ s⁻¹)** | 4964 | 4232 | 3404 |
|  | **C fluxes (GgC/month)** | **C fluxes (GgC/month)** | **C fluxes (GgC/year)** |
| **DOC load** | $27 \pm 5$ | $24 \pm 6$ | $307 \pm 68$ |
| **POC load** | $38 \pm 26$ | $12 \pm 5$ | $301 \pm 182$ |
| **DIC load** | $48 \pm 7$ | $28 \pm 8$ | $455 \pm 92$ |
| **CO₂ emissions** | $32 \pm 7$ | $35 \pm 6$ | $402 \pm 82$ |

**Table 4: $pCO_2$, pH and $CO_2$ evasion of several Southeast Asian rivers flowing into the South China Sea. *The sampling points were located outside of the peat area, so the actual peat coverage at that point was zero.**

| | Country | Catchment size (km²) | Discharge (m³ s⁻¹) | Peat coverage (%) | pH | $pCO_2$ (µatm) | $CO_2$ evasion (g m⁻² d⁻¹) | Reference |
|---|---|---|---|---|---|---|---|---|
| **Mekong** | Vietnam/Myanmar/Laos/Thailand/Cambodia | 795,000 | 15,000 | - | 7.4-7.9 | 1090 | 2.3 | Li et al., 2013 |
| **Xijiang** | China | 350,000 | 7,290 | - | 7.6 ± 0.2 | 2600 | 2.2-4.2 | Yao et al., 2007 |
| *Rajang* | *Malaysia* | *51,500* | *3,300* | *11* | *6.8 ± 0.1* | *2825 ± 508* | *2.4 ± 0.5* | *This study* |
| **Musi** | Indonesia | 56,931 | 3,050 | 3.5 | 6.9 ± 0.3 | 4316 ± 928 | 7.6 ± 3.2 | Wit et al., 2015 |
| **Red** | China/Vietnam/Laos | 156,450 | 2,640 | - | 8.1 | 1589 ± 885 | 6.4 ± 0.2 | Le et al., 2018 |
| **Batang Hari** | Indonesia | 44,890 | 2,560 | 5 | 7.1 | 2400 ± 18 | 3.9 ± 0.8 | Wit et al., 2015 |
| **Indragiri** | Indonesia | 17,968 | 1,180 | 11.9 | 6.3 ± 0.1 | 5777 ± 527 | 10.2 ± 2.7 | Wit et al., 2015 |
| **Siak** | Indonesia | 10,423 | 684 | 21.9 | 5.1 ± 0.5 | 8555 ± 528 | 14.1 ± 2.7 | Wit et al., 2015 |
| **Lupar** | Malaysia | 6,558 | 490 | 30.5* | 6.9 ± 0.3 | 1274 ± 148 | 2.0 ± 0.5 | Müller et al., 2016 |
| **Saribas** | Malaysia | 1,943 | 160 | 35.5* | 7.3 | 1159 ± 29 | 1.1 ± 0.9 | Müller et al., 2016 |
| **Maludam** | Malaysia | 91 | 4 | 100 | 3.8 ± 0.2 | 8100 ± 500 | 9.1 ± 4.7 | Müller et al., 2015 |