# Peer review of "Impact of peatlands on carbon dioxide (CO2) emissions from the Rajang River and Estuary, Malaysia"

_Biogeosciences, 2018_

## Referee Comment (RC1) · Anonymous Referee #1 · 3 Oct 2018

The manuscript (MS) submitted by Müller-Dum et al. investigates the C exports from the Rajang River and Estuary (Indonesia) based on sampling cruises during wet and dry season. That includes observations of CO2 partial pressures (pCO2), calculation of CO2 emissions from the water surface, and lateral exports of DOC, POC, and DIC. pCO2 and emissions are detailed for the peat-draining, non-peat-draining and estuarine parts of the river. One important result is that although the peat cover in the basin is significant, its contribution to C exports from the river system is not visible, as the peatlands are concentrated around the river delta. The manuscript of Müller-Dum et al. is of interest for the readership of Biogeosciences, because it reports the first pCO2 and CO2 emission estimates of this important river in SE-Asia, which is surprisingly

different from what would have been expected from observation from over peat draining rivers in this area. The methodology is well described and seems to be sound. The MS is in most parts well written. The results support the main conclusions drawn in the MS. The discussion of results is thorough and covers well the state of the art with respect to literature references. I suggest the publication of the MS after some moderate revisions. Please, find my comments to the authors below.

Major comment: You have been measuring pCO2 for quite different parts of the delta system delta (estuary and peat part of the river network) during the wet and the dry season. That becomes quite apparent from the figure 4. Did you do anything to compensate for the discrepancy in observed delta parts? If not, I would suggest that you calculate and report the average wet and dry season pCO2 only for the parts you have been sampling in both seasons.

General comments: Abstract: The abstract is comprehensible and summarizes well the main findings. However, the abstract would need some minor restructuring: P2, L8-9: It's not easy to see here how these DIC and delta13C values show that peatlands are not the main source. That would require some more explanation within the abstract. Maybe you could discard these two number from the abstract.

P2, L10: This sentence is repeating what was stated two sentences before.

P2, L10-12: "Thus...". I feel this sentence should conclude the abstract.

P2, L13-15: "CO2 fluxes...". This sentence should come slightly earlier and directly follow your statements related to the pCO2.

Introduction: P3, L2-3: Make clear that you are talking about terrestrial derived C fluxes.

P3, L13-14: Could you report the proportion of the water flux for comparison?

P3, L25-26: Did you do longitudinal transects from no-peat-influenced river reaches to river reaches surrounded by peat? If yes, it would be good to state that here.

P3, L26-27: Maybe you should discard that last sentence.

Methodology: The only thing I miss is an explanation why you observed the delta13C of DIC, and maybe the endmembers you used for your isotopic mixing model, if you applied one.

Results P9,L5-12: With regard to the positive correlation between delta13C and DIC concentration in the estuary: What is the marine endmember of delta 13C in DIC here? With regard to the negative correlation between delta13C and DIC concentration in the freshwater part: Is that correlation even stronger between delta13C and pCO2?

P9, L8: "Calculate DIC for the wet ...". For which part of the river network? The freshwater part? Please, clarify!

P9, L21-13: Is it possible to distinguish pCO2 observations you made during high, rising, falling, and low tide during your cruises? Or were your cruises in the delta predominantly done during a specific part of the tidal cycle? Were those different for wet and dry season cruises?

P9, L27-28: Does that mean you cannot distinguish the diurnal variations from tidal variations for the delta? And you do not have enough data from the non-tidal part to identify a diurnal signal? Please, clarify.

P9, L30 – P10,L1: How did you calculate those gas exchange velocities? I see how your calculations compare well to the A11 model, but R01 model seems to be quite far off. Are those the results for the whole river system?

P10, L3-5: Those emission rates refer to the entire observed river network? Did you weight the emission rates along the longitudinal profile by stream width?

---

## Referee Comment (RC2) · Anonymous Referee #2 · 30 Oct 2018

Comments to the manuscript by Mueller-Dum et al., "Impact pf peatlands on carbon dioxide (CO2) emissions from the Rajang River and Estuary, Malaysia".

General comment: The manuscript focus on an important topic that I believe is suitable for publication in Biogeosciences. The transport and emission of carbon/GHG′s from river networks has repeatedly been concluded during the last decade as a highly significant component when for example estimating landscape C budgets at various scales and biomes. Although the importance is well-recognized, I would claim that relatively little is known about large rivers and their source contribution of atmospheric CO2. The knowledge that exists is largely restricted by the spatiotemporal resolution of the mea-

surements or by using data being based on indirect measurements of pCO2. There is also a clear bias in existing data-sets towards northern hemisphere river networks and with limited information of tropical rivers, especially south-east Asian ones. In this context this study aims to fill an important gap in our understanding concerning large scale drivers of aquatic C in river networks. The influence of peat deposits in the catchment on the pGHG in the water has been shown for various biomes and river network sizes but more extensive investigations are needed. Hence, this is a highly relevant topic especially for a tropical region like this.

Although the aim of manuscript is important I have some concerns on how suitable the manuscript is for publication in its current form. My main concerns are: 1) How the actual emissions are calculated. I understand that this is a data scarce region but the way the authors have estimated the emissions is not especially convincing. The author's measure pCO2 in a satisfactory way but the entire k calculation component feels very shaky. No actual measurements of any of the input parameters are conducted. A vague estimate of a fixed water velocity is used in combination with modelled wind data. Three different k parameterizations are then used gaining slightly, to very, different outputs. The model producing intermediate k estimates are then used without any stronger further motivation. The whole procedure feels as I already said very shaky, without knowing anything about the river, investigating seasonal differences in emissions and then using a fixed water velocity sounds for example very strange. On top of these vague calculation steps there are no uncertainty estimate of the calculated emissions (or lateral exports of inorganic and organic C!!). To describe and estimate this in a transparent way would be a requirement in my eyes, especially due to the scarcity in data for the k calculations. If this is problematic to handle, one suggestion is to skip the emission data and solely present the pCO2 patterns and how it varies with wet and dry season and the influence of peatlands. Personally I think this would be the way to go and would be highly interesting in itself. 2) I am not totally convinced of the interpretations of the 13C-DIC data, I am surprised by the generally high 13C-DIC values, the authors claim that the contribution by carbonate containing bedrock to the

riverine DIC is minimal in the area and that the river is affected by tidal water sustaining the estuary with marine DIC. That is likely correct but the high 13C-DIC is found even in upstream non-peat area, is the evasion the sole explanation for that? Maybe not relevant, but what about methane production, I understand that methane might have been included in the original plan, but if methane in the peatlands is mainly produced by CO2 reduction this will heavily influence the 13C of the CO2 being delivered to the river (See Campeau et al. 2018 for example). Overall, I find the interpretation of the 13C-DIC data quite short and not as well developed as it could be. 3) Is it really correct to talk about seasonality when just two measurement campaigns are conducted, i.e. wet and dry season? I am not familiar with the region but to call something seasonality or similar would in my mind require a higher sampling resolution in time.

Detailed comments:

P3 Ln 1-10, there is a mix of wetland and peatland, consistency or a clear separation would be good.

P3 Ln 11, Odd formulation and scientifically a bit weird. To claim that something is the highest worldwide is only true until someone else present a higher number. I would recommend to be more open in this formulation.

P5 Ln 20-25 and 30, what about correction for salinity on the pCO2 and emissions?

P6 Ln 8-10 Isn't water velocity dependent on discharge, why is a fixed value used???

P6 Ln 10, Is there no wind data to validate this modeled data with? How accurate is the wind data compared to conditions over the river is tricky to judge. Feels very vague and uncertain!!!

Also, how was water depth measured, it is not mentioned as far as I see, but included in the B04.

Based on the fixed water velocity and fixed wind?? Is a constant k used for each season independent of location along the river?

P7 Ln 29-30, a bit odd that POC was measured but not DOC. Hard to redo the study but how relevant are the literature DOC values for this study, please motivate better!

P8 Ln 25, please clarify what pH that is for wet resp. dry season.

P9 Ln 10-12, was not the purpose to investigate if the peatlands have an influence on the pCO2 in the river. Feels a bit strange then to say that too few 13C-DIC samples were taken.

P9 Ln 20, here and elsewhere, what is "distributaries", isn't just tributaries enough???

P9 Ln 21-23, important sentence but feels more like discussion than result!!

P9 Ln 27-28, again, feels more like discussion to me.

P10 L4, what does the +-0.52 and +-0.45 mean? Some kind of uncertainty or just spread? Please clarify in the methods. The emission rates (and lateral exports of C) are hard to get a feeling of, how uncertain are they? Impossible to judge for the moment.

P10 Ln 17-20, Feels from a reader perspective a bit odd to start to say that the findings are the same as found in other studies. I think the authors could "sell" their study better than that. It is important information but I would not place it first in the discussion. Also, maybe a matter of personal taste, but why not start with the main focus of the manuscript in the discussion (pCO2 patterns and maybe emissions if included), the SPM and POC story is secondary as I see it.

P12 Ln 11-14, Likely true but there is also a strong fractionation in 13C-DIC related to changes/differences in pH which could be up to ca 10 per mille.

Table 2. What is the +- of the emissions, the SE of the mean? I.e. some kind of measure of the spatial variability? Is this driven by something else than just variability in pCO2? Is k fixed for all data? According to the methods I get this feeling. Please clarify in the methods.

References: Campeau, A., Bishop, K., Nilsson, M. B., Klemedtsson, L., Laudon, H., Leith, F. I., Öquist, M. G., Wallin, M. B., 2018. Stable carbon isotopes reveal soil-stream DIC linkages in contrasting headwater catchments, Journal of Geophysical Research – Biogeosciences, 123 (1), 149-167, doi:10.1002/2017JG004083

---

## Author Comment (AC1) · 9 Nov 2018

We would like to thank you for taking the time to review our manuscript and for the overall positive evaluation. Below, we detailed how we are going to address your specific comments.

*The manuscript (MS) submitted by Müller-Dum et al. investigates the C exports from the Rajang River and Estuary (Indonesia) based on sampling cruises during wet and dry season. That includes observations of CO2 partial pressures (pCO2), calculation of CO2 emissions from the water surface, and lateral exports of DOC, POC, and DIC. pCO2 and emissions are detailed for the peat-draining, non-peat-draining and estuar-*

[Figure]

*ine parts of the river. One important result is that although the peat cover in the basin is significant, its contribution to C exports from the river system is not visible, as the peatlands are concentrated around the river delta. The manuscript of Müller-Dum et al. is of interest for the readership of Biogeosciences, because it reports the first pCO2 and CO2 emission estimates of this important river in SE-Asia, which is surprisingly different from what would have been expected from observation from over peat draining rivers in this area. The methodology is well described and seems to be sound. The MS is in most parts well written. The results support the main conclusions drawn in the MS. The discussion of results is thorough and covers well the state of the art with respect to literature references. I suggest the publication of the MS after some moderate revisions. Please, find my comments to the authors below.*

*Major comment: You have been measuring pCO2 for quite different parts of the delta system delta (estuary and peat part of the river network) during the wet and the dry season. That becomes quite apparent from the figure 4. Did you do anything to compensate for the discrepancy in observed delta parts? If not, I would suggest that you calculate and report the average wet and dry season pCO2 only for the parts you have been sampling in both seasons.*

We agree that in order to compare data from two seasons, we have to make sure to compare data from the same geographical extent. This was done for the non-peat reaches of the river, as they extended from Kapit to Kanowit during both seasons. The peat reaches differed geographically, due to the different salt intrusion limits during wet and dry season. For the estuary reaches, we reported the averages for the two different seasons accompanied by their salinity and the remark that different locations were sampled. However, we see how this might be confusing. Therefore, in the revised paper, we will try to make the comparison more precise by comparing distinct parts of the river: non-peat (Kapit-Kanowit), peat that was non-saline during both seasons (Kanowit-Sibu), and the delta. This way, at least peat and non-peat areas are directly comparable because they geographically overlap. We think that using these three "new" categories does not only make more sense, it also improves the readability

of the manuscript, since only three categories are used (non-peat, peat, delta) instead of five (which were partially overlapping: non-peat, peat, estuary, delta, freshwater).

At the same time, we would take your comment into account about weighting emissions by stream width. This is now easily possible, as a recent publication by Allen and Pavelsky (2018) provides us with stream widths for the entire river. With those at hand, we are able to calculate $CO_2$ emissions for each category separately using specific stream widths. In addition, we will estimate the fraction of the catchment that belongs to each category and calculate area-weighted means of the measured parameters (SPM, POC, $pCO_2$, DO...).

In Summary, we suggest the following main changes to the manuscript:

- Definition of the three categories (their characteristics like water surface area, catchment fraction etc. will be summarized in an own Table)

- Instead of reporting freshwater averages and then peat/non-peat averages, values will be reported for the three categories peat, non-peat, delta, and in addition, an area-weighted mean will be calculated for peat and non-peat area combined. This way, we could merge what are currently Table 1 and 2.

- River loads will be recalculated using the new categories.

- The estimate of river surface area will be improved by using the now available GRWL Database.

- The estimate of catchment area will be improved and catchment fractions will be calculated for the three categories.

Further minor changes will be made as detailed below.

*General comments:*
*Abstract: The abstract is comprehensible and summarizes well the main findings. However, the abstract would need some minor restructuring:*
*P2, L8-9: It's not easy to see here how these DIC and delta13C values show that peatlands are not the main source. That would require some more explanation within the abstract. Maybe you could discard these two number from the abstract.*
Agreed, we will delete the sentence about DIC and delta13C.

*P2, L10: This sentence is repeating what was stated two sentences before.*
We will delete this sentence in the revised manuscript.

*P2, L10-12: "Thus: : :". I feel this sentence should conclude the abstract.*
We will move this sentence to the end of the abstract.

*P2, L13-15: "CO2 fluxes: : :". This sentence should come slightly earlier and directly follow your statements related to the pCO2.*
We will move this sentence up, so that the statement about the $CO_2$ fluxes directly follows the statement about $pCO_2$ values.

*Introduction:*
*P3, L2-3: Make clear that you are talking about terrestrial derived C fluxes.*
We will change the first sentence to: "Tropical rivers transport large amounts of terrestrially derived carbon to the ocean and the atmosphere (Aufdenkampe et al., 2011; Raymond et al., 2013)."

*P3, L13-14: Could you report the proportion of the water flux for comparison?*
We will add the following information: "Because of these high DOC concentrations, Indonesian rivers may account for 75 % of the DOC flux into the South China Sea (SCS) while accounting for 39 % of the discharge (Huang et al., 2017)."

*P3, L25-26: Did you do longitudinal transects from no-peat-influenced river reaches to river reaches surrounded by peat? If yes, it would be good to state that here.*
Yes, we will add a sentence for clarification: "To this end, we surveyed longitudinal transects from river reaches that were not influenced by peat to the peat-covered delta."

*P3, L26-27: Maybe you should discard that last sentence.*

The sentence "We expected to see a clear peat signal, i.e. elevated $CO_2$ concentrations in the peat area" will be deleted.

*Methodology:*
*The only thing I miss is an explanation why you observed the delta13C of DIC, and maybe the endmembers you used for your isotopic mixing model, if you applied one.*
We will add the following justification: "In August 2016, water samples were also taken for the determination of dissolved inorganic carbon (DIC) and the isotopic composition ($\delta^{13}$C) of DIC, because the isotopic composition of DIC can help in identifying its sources." An isotopic mixing model was not applied.

*Results*
*P9,L5-12: With regard to the positive correlation between delta13C and DIC concentration in the estuary: What is the marine endmember of delta 13C in DIC here?*
We measured a marine end-member at one station off the Rajang River mouth during our survey in August 2016 (DIC = 2347 $\mu$mol/L, d13C-DIC = -1.97 $^0/_{00}$). This value (for DIC) was used for the $CO_2$ mixing model as described in the Supplement. If you mean a calculated (theoretical) end-member, it depends on the DIC value that we consider marine, and that, in turn, depends on the salinity that we consider the marine end-member to have. In this part of the South China Sea, a salinity of 33 can be considered marine. The calculated DIC end-member would be 2310 $\mu$mol/L at this salinity, and the calculated d13C-DIC would be -2.96 $^0/_{00}$. We think that using the measured end-member is more appropriate, as done in the $CO_2$ mixing model. However, since we are not applying an isotopic mixing model, we found it unnecessary to mention these numbers in the text. However, in the revised manuscript, we will add a Keeling plot showing the regression line for freshwater samples, where the y-axis intercept should give the source d13C-DIC.

*With regard to the negative correlation between delta13C and DIC concentration in the freshwater part: Is that correlation even stronger between delta13C and pCO2?*
No, it is weaker. This can partially be explained by the sampling: $\delta13$C and DIC were

analyzed from the same discrete water sample. $CO_2$, in contrast, was monitored with the equilibrator, so taken from a slightly different spot (the water sampler and the water intake for the equilibrator were in the front and rear of the boat, respectively).

*P9, L8: "Calculate DIC for the wet : : :". For which part of the river network? The freshwater part? Please, clarify!*
Yes, this value states the calculated DIC for the freshwater part of the river. In the revised manuscript, this sentence will have to be changed due to the new three categories.

*P9, L21-13: Is it possible to distinguish pCO2 observations you made during high, rising, falling, and low tide during your cruises? Or were your cruises in the delta predominantly done during a specific part of the tidal cycle? Were those different for wet and dry season cruises?*
Tidal variability of $pCO_2$ was only observed at the river mouth. In large parts of the delta, $pCO_2$ was relatively uniform, as can be seen in the Figure below. The only information about tidal variability was therefore obtained from the stationary measurement in Sarikei at the Rajang River mouth, where tidal variability is quite substantial for both water level (up to 5 m) and $pCO_2$ (between 2000 and 6000 $\mu$atm). We will show the time series of $pCO_2$ and the water level for both January and August in the Supplement of the revised manuscript. Please see Fig.1 below.

*P9, L27-28: Does that mean you cannot distinguish the diurnal variations from tidal variations for the delta? And you do not have enough data from the non-tidal part to identify a diurnal signal? Please, clarify.*
This is both correct. In the tidal part, we cannot distinguish tidal and diurnal signals, because we essentially have one stationary measurement that was conducted overnight. This measurement (will be shown in the Supplement, see comment above) suggests that the variation with tide dominates over the diurnal variation. In the non-tidal part, we do not have enough night-time data to identify a diurnal signal at all. We will add the following explanations: "Unfortunately, our data did not allow to identify a diurnal signal

of either $pCO_2$ or DO. In the tidal part of the river, we have only the one stationary measurement overnight, when a diurnal signal could not be identified due to the strong tidal signal. In the non-tidal part of the river, we do not have enough night-time data in order to make a sound statement about a difference between day- and night time $pCO_2$ and DO."

*P9, L30 – P10,L1: How did you calculate those gas exchange velocities? I see how your calculations compare well to the A11 model, but R01 model seems to be quite far off. Are those the results for the whole river system?*
The calculation of gas exchange velocities is detailed in Section 2.3 and results are compared in the Supplement. When we report our main results here, the used parameterization for the calculation of the gas exchange velocity is stated in parentheses (k600, B04). We will add "using the B04 model" in this sentence to make it more obvious to the reader which model was used. It was also stated in the following that "Fluxes reported in this study are calculated from the B04 model, which yielded intermediate values. It was chosen because it recognizes flow velocity as a driver of turbulence in addition to wind speed. Results for the other two models are compared in the Supplement." Those results are for the whole river system. The B04 model was constructed for estuaries based on measured relationship with wind speed and a parameterization for streams with current velocity as driver from the literature, thus seems appropriate for both river and estuarine reaches. As for A11, the river reaches that we studied ranged in width from 271m to several kilometers in the delta, thus meet the criteria for a "large river" defined by the authors (>100 m width). R01 used data from several rivers and estuaries to derive their predictive equation, among which are different rivers and estuaries for which the width is not given. After receiving your comments, we looked into it and found that the R01 rivers and estuaries probably range in width from 600m to several kilometers. This could be the reason why R01 differs so substantially from the other two models. However, R01 was used in our manuscript only for comparison, to show the range of possible values with implications for the uncertainty of emission estimates (which is always high when k is not directly measured). In the

revised manuscript, we will present all three emission estimates as equally valid and present the range of values that they give instead of picking one of the models. This is due to a major comment of Referee 2, who suggested to improve the discussion of uncertainties that arise from the k models.

*P10, L3-5: Those emission rates refer to the entire observed river network? Did you weight the emission rates along the longitudinal profile by stream width?*
Those emission rates were calculated for the freshwater part of the river. They were not weighted by stream width, because the values represented emission rates per water surface unit area (gC/m2/d). In the revised manuscript, we will calculate those emission rates separately for peat, non-peat area and for the delta and calculated emissions (g/month) using the water surface area in each of the categories.

References
Allen, G. H. and Pavelsky, T. M. Global extent of rivers and streams. Science 361, 585-588, doi: 10.1126/science.aat0636, 2018.

[Figure]

**Fig. 1.**

---

## Author Comment (AC2) · 10 Nov 2018

Thank you for taking the time to review our manuscript and for your comments and suggestions. We believe that we can satisfactorily address each of your comments.

*General comment: The manuscript focus on an important topic that I believe is suitable for publication in Biogeosciences. The transport and emission of carbon/GHGs from river networks has repeatedly been concluded during the last decade as a highly significant component when for example estimating landscape C budgets at various scales and biomes. Although the importance is well-recognized, I would claim that relatively little is known about large rivers and their source contribution of atmospheric*

[Figure]

*CO2. The knowledge that exists is largely restricted by the spatiotemporal resolution of the measurements or by using data being based on indirect measurements of pCO2. There is also a clear bias in existing data-sets towards northern hemisphere river networks and with limited information of tropical rivers, especially south-east Asian ones. In this context this study aims to fill an important gap in our understanding concerning large scale drivers of aquatic C in river networks. The influence of peat deposits in the catchment on the pGHG in the water has been shown for various biomes and river network sizes but more extensive investigations are needed. Hence, this is a highly relevant topic especially for a tropical region like this.*

*Although the aim of manuscript is important I have some concerns on how suitable the manuscript is for publication in its current form. My main concerns are: 1) How the actual emissions are calculated. I understand that this is a data scarce region but the way the authors have estimated the emissions is not especially convincing. The author's measure pCO2 in a satisfactory way but the entire k calculation component feels very shaky. No actual measurements of any of the input parameters are conducted. A vague estimate of a fixed water velocity is used in combination with modelled wind data. Three different k parameterizations are then used gaining slightly, to very, different outputs. The model producing intermediate k estimates are then used without any stronger further motivation. The whole procedure feels as I already said very shaky, without knowing anything about the river, investigating seasonal differences in emissions and then using a fixed water velocity sounds for example very strange. On top of these vague calculation steps there are no uncertainty estimate of the calculated emissions (or lateral exports of inorganic and organic C!!). To describe and estimate this in a transparent way would be a requirement in my eyes, especially due to the scarcity in data for the k calculations. If this is problematic to handle, one suggestion is to skip the emission data and solely present the pCO2 patterns and how it varies with wet and dry season and the influence of peatlands. Personally I think this would be the way to go and would be highly interesting in itself. 2) I am not totally*

[Figure]

*convinced of the interpretations of the 13C-DIC data, I am surprised by the generally high 13C-DIC values, the authors claim that the contribution by carbonate containing bedrock to the riverine DIC is minimal in the area and that the river is affected by tidal water sustaining the estuary with marine DIC. That is likely correct but the high 13C-DIC is found even in upstream non-peat area, is the evasion the sole explanation for that? Maybe not relevant, but what about methane production, I understand that methane might have been included in the original plan, but if methane in the peatlands is mainly produced by CO2 reduction this will heavily influence the 13C of the CO2 being delivered to the river (See Campeau et al. 2018 for example). Overall, I find the interpretation of the 13C-DIC data quite short and not as well developed as it could be. 3) Is it really correct to talk about seasonality when just two measurement campaigns are conducted, i.e. wet and dry season? I am not familiar with the region but to call something seasonality or similar would in my mind require a higher sampling resolution in time.*

Thank you for highlighting the relevance of our research and for suggesting improvements. We would like to respond to each of your main concerns first. Below, we will respond to each of your minor comments.

1) We agree that k is the most uncertain parameter in $CO_2$ emission calculations. It is very common in the scientific literature to use one of the available k-parameterizations to calculate $CO_2$ emissions, and only in very few cases were authors able to provide both $pCO_2$ measurements and flux measurements at the same time. Of course we understand that just because something is usually done in a certain way, it doesn't mean it is also justified. We agree that the paper would also work without the $CO_2$ emission calculations and in principle, we are open to this modification. However, rather, we would keep the $CO_2$ emission estimates as part of our study and we have two main arguments for that:

The first one concerns the insights that a $CO_2$ emission estimate allows. We agree that it is necessary to better account for uncertainties, but keeping $CO_2$ emission estimates as part of our analysis allows us to set lateral transport in relation to $CO_2$ outgassing. This is important in the light of the "active pipe" hypothesis, which sees rivers as active conduits and locations of carbon processing and outgassing instead of mere transport pipes. We can only make a contribution to the validation or falsification of this hypothesis if we provide estimates for both lateral transport and vertical outgassing. Therefore, we would like to keep $CO_2$ emission estimates as part of our analysis.

However, we agree that uncertainties must be accounted for in a more transparent and suitable way. We think that singling out one of the k-parameterizations as the preferred one might be the main reason the whole procedure feels shaky. While we think that we do have some justification for that (Borges et al. 2004 were the only ones who considered flow velocity as driver of turbulence), we acknowledge that the uncertainty in k must be better accounted for. Therefore, we will present the different parameterizations that we found suitable for our river as equally justified and interpret the range of values that they yield as a range of uncertainty introduced by the decision to use a parameterization for k. In summary, this means: We will present all three parameterizations by reporting an average, minimum and maximum estimate. This way, it is easier for the readers to get an idea of the uncertainty.

The second argument is related to the reception of scientific evidence by the readership. There is, to our knowledge, only one other $CO_2$ emission estimate for the Rajang River (Chen et al., 2013). Those authors used a k-parameterization to calculate fluxes (Wanninkhof 1992) and $pCO_2$ data from one season (inter-monsoonal) only. Our data increase the data density and considering the importance of $CO_2$ emissions we find it important that all available data is at hand and accessible for the scientific community. Accessibility is a lot easier when we report data in matching ways, therefore, it would be important for us to provide a $CO_2$ flux estimate that can be compared to existing ones.

2) Our measurements only cover the lower river reaches (approx.. the last 200 km), so outgassing might indeed be a valid explanation for the high $\delta 13$C-DIC. Before the river reaches Kapit (which is the point up to which we have measurements), it flows through mountainous terrain, even including rapids, where high outgassing might occur. However, we will expand and further develop the discussion of $\delta 13$C-DIC in the revised manuscript, including a Keeling plot that should support the discussion of possible sources and also including processes that we had not mentioned so far, like methanogenesis.

3) A higher sampling frequency also during inter-monsoonal periods would certainly be desirable, just like interannual sampling. Our data from the peak of the wet and dry period is, of course, only a snapshot. The terminology "wet and dry season" still seems appropriate to us, as it describes accurately when the samples were taken. However, we agree that it is not possible to make strong claims about seasonality using this data. In the revised manuscript, we will make it clearer that "wet and dry season" is mainly a terminology and that our data are too few to make strong claims about seasonality. In Section 4.2.2, we would add a sentence: "As our data was collected during two single surveys, they represent only a snapshot and do not allow strong claims about seasonality."

*Detailed comments:*
*P3 Ln 1-10, there is a mix of wetland and peatland, consistency or a clear separation would be good.*
Agreed, Borges et al. (2015) consider different kinds of wetlands in their analysis, while the Wit et al. (2015) study focuses on a specific kind of wetland (peatland). We would try to rephrase and suggest the following change: "Two regional studies independently showed that the partial pressure of $CO_2$ ($pCO_2$) in rivers increases with increasing wetland coverage in the catchment. Borges et al. (2015) established a relationship

between wetland extent and pCO$_2$ for African rivers. Wit et al. (2015) presented an analog synthesis for Southeast Asian rivers, which flow through peatlands. Peatlands are a special type of wetland, where organic matter accumulates at rates that make them the most effective terrestrial carbon store on a millennial timescale (Dommain et al., 2011). Southeast Asian peatlands store 68.5 Gt carbon (Page et al., 2011). The highest riverine dissolved organic carbon (DOC) concentrations reported so far were found in Southeast Asian peat-draining rivers (Alkhatib et al. 2007; Moore et al., 2011; Müller et al., 2015), with an annual average of 68 mg L-1 DOC found in an undisturbed peat-draining river (Moore et al., 2013). Because of these high DOC concentrations, Indonesian rivers may account for 75 % of the DOC flux into the South China Sea (SCS) while accounting for 39 % of the discharge (Huang et al., 2017). Surprisingly, CO$_2$ emissions from these rivers are not exceptionally high (Müller et al., 2015; Wit et al., 2015). This is attributed to a short residence time of the organic matter in the river, allowing little time for decomposition, and the resistance of peat-derived carbon to bacterial degradation. Nevertheless, the CO$_2$ flux from peat-draining rivers to the atmosphere increases with increasing peat coverage in the river basin (Wit et al., 2015), showing that these ecosystems exert an important influence on a river's carbon budget."

*P3 Ln 11, Odd formulation and scientifically a bit weird. To claim that something is the highest worldwide is only true until someone else present a higher number. I would recommend to be more open in this formulation.*
We would rephrase: "The highest riverine dissolved organic carbon (DOC) concentrations reported so far were found in Southeast Asian peat-draining rivers (Alkhatib et al. 2007; Moore et al., 2011; Müller et al., 2015), with an annual average of 68 mg L-1 DOC found in an undisturbed peat-draining river (Moore et al., 2013)."

*P5 Ln 20-25 and 30, what about correction for salinity on the pCO2 and emissions?*
Correction to pCO$_2$ was not applied, because salinity impacts the solubility of the gas, not its partial pressure, which is a notional variable. The independence of partial pressure from salinity or water temperature is an asset when comparing $CO_2$ in rivers across seasons or from different locations. In contrast, $CO_2$ emissions to the atmosphere are dependent on salinity, because the actual concentration of a gas is used in the calculation of its flux to the atmosphere. This is accounted for in our calculations in the calculation of $CO_2$ solubility according to Weiss (1974) and the calculation of the Schmidt number according to Wanninkhof (1992), which is used to adapt diffusivity and thus the gas exchange velocity k to different salinity and temperature. The information how we calculated $k_{T,S}$ from $k_{600}$ was indeed missing in the current version of the manuscript and will be added to the revised paper.

*P6 Ln 8-10 Isn't water velocity dependent on discharge, why is a fixed value used???*
In our response, we are using:
Q = Discharge
w = Water velocity
k = gas transfer velocity
It is correct that w depends on Q. Raymond et al. (2012) found that w scales with $Q^{(0.29\pm0.01)}$. In the absence of w measurements during our study, we had to resort to a literature value. We felt like it made more sense to use a literature value from the same river instead of calculating w with an empirical equation that was developed for rivers in the United States. However, from the hydraulic equation of Raymond et al. (2012), we can still get an idea about how variable w might be. Our reasoning is as follows: Ling et al. (2017) report w = 1.1 m/s. From the description of their work we infer that this is an average of all measurements they carried out in August 2014 and January 2015. The measurements of Staub and Esterle (1993) were carried out in July and August 1992, and a range and the average value are given in their paper (w=0.7 m/s). As both those estimates seem equally valid, we did not single out one but used their average of w = 0.9 m/s as a general average flow velocity in the Rajang River. During the monsoon season in January, Q increases by 50% compared to the average value. According to the equation by Raymond et al. (2012), a 50% increase in Q would result in a 12% increase of w. If we assume this variability, w in the Rajang River would vary

with Q from 0.8 to 1.0 m/s or $0.9 \pm 0.1$ m/s. This would add an uncertainty of 4% to the Borges et al. (2004) k-value. However, the uncertainty introduced by the use of different k-models (A11, R01, B04) is much larger: A11 yields up to 30% higher fluxes than B04 (the intermediate), and R01 resulted in up to 70% lower values. This shows that the choice of model is the biggest source of uncertainty. Therefore, we suggest that we include the described error analysis in the Supplement, but stick to our plan of deriving the overall uncertainty of k from the presentation of the different models.

*P6 Ln 10, Is there no wind data to validate this modeled data with? How accurate is the wind data compared to conditions over the river is tricky to judge. Feels very vague and uncertain!!!*
Unfortunately, we were unable to obtain wind speed data measured on site. So the NOAA NCEP Reanalysis data was the best available option. This is a solution authors resort to if no on-site wind speed data is available (e.g., Bouillon et al., 2012; a number of Russian estuaries reported in Chen et al., 2013). We will point out the uncertainty introduced by the choice of wind data in the discussion of the $CO_2$ emission estimates.

*Also, how was water depth measured, it is not mentioned as far as I see, but included in the B04.*
That is correct, this information is missing. Depth was recorded at each station from the bottom sounder of the boat. This information will be added to the revised manuscript.

*Based on the fixed water velocity and fixed wind?? Is a constant k used for each season independent of location along the river?*
Yes, since we derived wind speed for an entire grid and used a literature value for the water flow velocity, we had no choice but to use one k for the entire river for each season. We agree that this is not 100% satisfactory and will include some more justification in the Methods section. We also think that using the new approach following your main comment, uncertainties might be better accounted for.

*P7 Ln 29-30, a bit odd that POC was measured but not DOC. Hard to redo the study*

*but how relevant are the literature DOC values for this study, please motivate better!*
In fact, DOC samples were taken but the values had to be discarded because contamination was suspected. The DOC measurements of Martin et al. (2018) were taken during three campaigns in 2017 and covered the Rajang delta downstream of Kanowit. The author provided us with the DOC values that he published in his 2018 study and we chose only those that were taken at zero salinity. We believe that these DOC values are the best available estimate for the Rajang River.

*P8 Ln 25, please clarify what pH that is for wet resp. dry season.*
This will be done in the revised manuscript.

*P9 Ln 10-12, was not the purpose to investigate if the peatlands have an influence on the pCO2 in the river. Feels a bit strange then to say that too few 13C-DIC samples were taken.*
Yes, the purpose was to investigate the impact of peatlands on $pCO_2$. After the first survey and looking at the data, we found that the measurement of additional parameters might be helpful, so DIC and $\delta13$C-DIC were measured during the second campaign. However, resources were limited so the number of samples is not sufficient to make a statement about statistical significance.

*P9 Ln 20, here and elsewhere, what is "distributaries", isn't just tributaries enough???*
To our understanding, tributaries are rivers that flow into the main river. "Distributary", in contrast, is a word that describes when a river branches off from the main river. Thus, "tributary" and "distributary" describe two different things. In our study, we use the term "distributary" because the Rajang River does not discharge through one river mouth, but splits up into several "arms" (or "distributaries") before discharging into the sea. We use the term "distributary" in accordance with other descriptions of the Rajang River system by Staub and Esterle (1993), Staub et al. (2000), Staub and Gastaldo (2003).

*P9 Ln 21-23, important sentence but feels more like discussion than result!!*

Agreed. The first half of the sentence is a result and will stay here, the second part of the sentence belongs to the discussion and will be moved to the right section.

*P9 Ln 27-28, again, feels more like discussion to me.*
We would like to keep this statement about night time measurements of $CO_2/O_2$ here in the Results section. We inserted it after stating a correlation between $CO_2$ and $O_2$. An immediate question a reader might have is, if $CO_2$ and $O_2$ co-vary, is that due to diurnal variability? Therefore, we would like to take this thought up and quickly clarify that we are unable to make a statement about diurnal variability. We are not providing any further discussion, just stating the fact that not enough data was available, which we feel is appropriately placed in the Results section.

*P10 L4, what does the +-0.52 and +-0.45 mean? Some kind of uncertainty or just spread? Please clarify in the methods. The emission rates (and lateral exports of C) are hard to get a feeling of, how uncertain are they? Impossible to judge for the moment.*
This is the spread of the data and comes from the spread of $pCO_2$. However, following your main comment (1), we will now report $CO_2$ fluxes differently as described above. The reported errors will be described in the Methods section 2.3.

*P10 Ln 17-20, Feels from a reader perspective a bit odd to start to say that the findings are the same as found in other studies. I think the authors could "sell" their study better than that. It is important information but I would not place it first in the discussion.*
We will rewrite this paragraph and not put those 3 lines first in the discussion. However, we would still like to start the discussion off by generally characterizing the Rajang River (see next comment) and placing it in the "bigger picture" before we start the detailed discussion of $pCO_2$.

*Also, maybe a matter of personal taste, but why not start with the main focus of the manuscript in the discussion (pCO2 patterns and maybe emissions if included), the SPM and POC story is secondary as I see it.*

We agree that the SPM and POC story is secondary. We had the choice between reporting and discussing SPM/POC first, last, or leaving it out completely. The last option seems inappropriate, as SPM and POC measurements were conducted, the data has good quality and might be interesting to many readers. We saw no reason to exclude it. About the position in the manuscript: We thought that if we lay out the $pCO_2$ discussion first and then add SPM/POC, the whole SPM/POC discussion might come across as an afterthought. In the end, we decided to provide the reader with a general characterization of the rather unfamiliar Rajang River before detailing our thoughts about $pCO_2$. This general characterization also includes the SPM and POC data. We agree that this might be a matter of personal taste, but we feel that the SPM and POC story should be part of the general characterization followed by the more detailed discussion of $pCO_2$. Therefore, we would like to keep the current order.

*P12 Ln 11-14, Likely true but there is also a strong fractionation in 13C-DIC related to changes/differences in pH which could be up to ca 10 per mille.*
As stated in our response to your three major concerns, we will expand and deepen the discussion of $\delta13$C-DIC and we thank you for providing another highly relevant reference that we had not considered so far. As for pH, we are reporting the isotopic signature of the entire DIC pool. For a fixed DIC, a change in pH would certainly influence the equilibrium fractionation within the carbonate pool, but it would not change the $\delta13$C-DIC. Of course, pH would influence $\delta13$C-DIC if DIC is added or removed from the system (as, e.g., in $CO_2$ evasion). We will point out the influence of pH in the expanded discussion of $\delta13$C-DIC.

*Table 2. What is the +- of the emissions, the SE of the mean? I.e. some kind of measure of the spatial variability? Is this driven by something else than just variability in pCO2? Is k fixed for all data? According to the methods I get this feeling. Please clarify in the methods.*
Yes, the $\pm$ is the SE of the mean. This is stated in the caption (mean $\pm$ SE). The abbreviation "SE" is introduced in the caption of Table 1. As we present averages from

our longitudinal surveys, the SE is a measure of the spatial variability. For $pCO_2$ and $FCO_2$, it is driven by the variability of $pCO_2$. For $O_2$ and DIC, it is driven by the spatial variability of these parameters, respectively. As described above, for k one value for the wet and dry season was used, but we intend to provide a range instead in the revised manuscript. We would clarify in the Methods section and provide the range of k and $FCO_2$ values in the results table to make it clearer.

References

Bouillon, S., Yambele, A., Spencer, R. G. M., Gilikin, D. P., Hernes, P J., Six, J., Merckx, R., and Borges, A. V. Organic matter sources, fluxes and greenhouse gas exchange in the Oubangui River (Congo River basin). Biogeosciences 9, 2045-2062. doi: 10.5194/bg-9-2045-2012, 2012.

Chen, C.-T. A., Huang, T.-H., Chen, Y.-C., Bai, Y., He, X. and Kang, Y. Air-sea exchanges of CO2 in the world's coastal seas. Biogeosciences 10: 6509-6544. doi: 10.5194/bg-10-6509-2013, 2013.

Staub, J. R. and Esterle, J. S. Provenance and sediment dispersal in the Rajang River delta/ coastal plain system, Sarawak, East Malaysia. Sedimentary Geology 85: 191-201, 1993.

Staub, J. R. and Gastaldo, R. A. Late Quarternary Sedimentation and peat development in the Rajang River Delta, Sarawak, East Malaysia. In: F. Hasan Sidi, Nummedal, D., Imbert, P., Darman, H., and Posamentier, H. W. Tropical Deltas of Southeast Asia – Sedimentology, Stratigraphy, and Petroleum Geology. SEPM Special Publication No. 76, p. 71-87, Tulsa, Oklahoma, USA, September 2003.

Staub, J. R., Among, H. L., and Gastaldo, R. A. Seasonal sediment transport and deposition in the Rajang River delta, Sarawak, East Malaysia. Sediment Geol 133: 249-264, 2000.

---

## Author Response (AR1)

**Main changes in the revised manuscript include:**

1. Definition of three distinct and comparable categories: peat, non-peat, delta (their characteristics like water surface area, catchment fraction etc. are summarized in Table 1)
2. Instead of reporting freshwater averages and then peat/non-peat averages, values are reported for the three categories peat, non-peat, delta, and in addition, an area-weighted mean was calculated for peat and non-peat area combined (Table 2)
3. CO2 fluxes were calculated by comparing three different k-parameterizations. Instead of picking one of these estimates, a range of values (average, minimum, maximum) is given for CO2 fluxes and derived parameters
4. River loads were recalculated according to point 2 & 3
5. River surface area was recalculated using the GRWL Database.
6. The "non-peat contribution" calculation was removed. After changing the CO2 flux calculation, the results had such large uncertainties that we were unable to derive a statement from this calculation. Thus, it seemed pointless and was removed.

**Point-by-point response to the Referee's comments**

In the following, we present a point-by-point response to the reviews. We will keep it short by only outlining the changes we made with regard to those comments. For further justification or more detailed explanations, please see the author's response that we posted on Biogeosciences Discussions.

**Comments by Reviewer #1**

*The manuscript (MS) submitted by Müller-Dum et al. investigates the C exports from the Rajang River and Estuary (Indonesia) based on sampling cruises during wet and dry season. That includes observations of CO2 partial pressures (pCO2), calculation of CO2 emissions from the water surface, and lateral exports of DOC, POC, and DIC. pCO2 and emissions are detailed for the peat-draining, non-peat-draining and estuarine parts of the river. One important result is that although the peat cover in the basin is significant, its contribution to C exports from the river system is not visible, as the peatlands are concentrated around the river delta. The manuscript of Müller-Dum et al. is of interest for the readership of Biogeosciences, because it reports the first pCO2 and CO2 emission estimates of this important river in SE-Asia, which is surprisingly different from what would have been expected from observation from over peat draining rivers in this area. The methodology is well described and seems to be sound. The MS is in most parts well written. The results support the main conclusions drawn in the MS. The discussion of results is thorough and covers well the state of the art with respect to literature references. I suggest the publication of the MS after some moderate revisions. Please, find my comments to the authors below.*

*Major comment: You have been measuring pCO2 for quite different parts of the delta system delta (estuary and peat part of the river network) during the wet and the dry season. That becomes quite apparent from the figure 4. Did you do anything to compensate for the discrepancy in observed delta parts? If not, I would suggest that you calculate and report the average wet and dry season pCO2 only for the parts you have been sampling in both seasons.*

In the revised manuscript, we introduce three distinct categories: peat, non-peat and delta. Observations in the peat and non-peat areas are directly comparable between seasons. They were defined such that they were non-saline and covered by our observations during both seasons. Delta values are reported for the sake of completeness, we made clear in the Methods section that they are not directly comparable between seasons.

The introduction of three new categories required a recalculation of averages. Instead of reporting freshwater averages and then peat/non-peat values, we report averages for peat, non-peat and delta and we calculated an area-weighted mean for peat/non-peat (Table 2). Following this new approach, we recalculated river loads and also emissions to the atmosphere. River surface area was recalculated using the GRWL database.

*General comments:*
*Abstract: The abstract is comprehensible and summarizes well the main findings. However, the abstract would need some minor restructuring:*
*P2, L8-9: It's not easy to see here how these DIC and delta13C values show that peat-lands are not the main source. That would require some more explanation within the abstract. Maybe you could discard these two number from the abstract.*
This sentence was deleted.

*P2, L10: This sentence is repeating what was stated two sentences before.*
This sentence was deleted.

*P2, L10-12: "Thus: : :". I feel this sentence should conclude the abstract.*
This sentence now concludes the abstract.

*P2, L13-15: "CO2 fluxes: : :". This sentence should come slightly earlier and directly follow your statements related to the pCO2.*
The statement about the $CO_2$ fluxes now directly follows the statement about $pCO_2$ values.

*Introduction:*
*P3, L2-3: Make clear that you are talking about terrestrial derived C fluxes.*
Sentence was changed to: "Tropical rivers transport large amounts of terrestrially derived carbon to the ocean and the atmosphere (Aufdenkampe et al., 2011; Raymond et al., 2013)."

*P3, L13-14: Could you report the proportion of the water flux for comparison?*
We added the following information: "Because of these high DOC concentrations, Indonesian rivers may account for 75 % of the DOC flux into the South China Sea (SCS) while accounting for 39 % of the discharge (Huang et al., 2017)."

*P3, L25-26: Did you do longitudinal transects from no-peat-influenced river reaches to river reaches surrounded by peat? If yes, it would be good to state that here.*

We added a sentence for clarification: "To this end, we surveyed longitudinal transects extending from river reaches that were not influenced by peat to the peat-covered delta."

*P3, L26-27: Maybe you should discard that last sentence.*
It was deleted.

*Methodology:*
*The only thing I miss is an explanation why you observed the delta13C of DIC, and maybe the endmembers you used for your isotopic mixing model, if you applied one.*
We added the following justification: "In August 2016, water samples were also taken for the determination of dissolved inorganic carbon (DIC) and the isotopic composition ($\delta^{13}C$) of DIC, because the isotopic composition of DIC can help in identifying its sources (Das et al., 2005; Campeau et al., 2017; 2018)." An isotopic mixing model was not applied.

*Results*
*P9,L5-12: With regard to the positive correlation between delta13C and DIC concentration in the estuary: What is the marine endmember of delta 13C in DIC here?*
Please see the reply to this question in our Author Comment. No changes were made with regards to this comment.

*With regard to the negative correlation between delta13C and DIC concentration in the freshwater part: Is that correlation even stronger between delta13C and pCO2?*
Please see the reply to this question in our Author comment. No changes were made with regard to this comment.

*P9, L8: "Calculate DIC for the wet : : :". For which part of the river network? The freshwater part? Please, clarify!*
This now reads: "Calculated DIC for the wet season averaged 289.8 ± 32.1 µmol $L^{-1}$ (area-weighted mean for the non-peat and peat area)."

*P9, L21-13: Is it possible to distinguish pCO2 observations you made during high, rising, falling, and low tide during your cruises? Or were your cruises in the delta predominantly done during a specific part of the tidal cycle? Were those different for wet and dry season cruises?*
Tidal variability is now discussed in Section 3.2 and 4.2.3 and time series of both pCO2 and water level are shown in the revised Supplement.

*P9, L27-28: Does that mean you cannot distinguish the diurnal variations from tidal variations for the delta? And you do not have enough data from the non-tidal part to identify a diurnal signal? Please, clarify.*
We added the following explanation: "Unfortunately, our data did not allow identification of a diurnal signal for either $pCO_2$ or DO. In the tidal part of the river, we had only the one stationary measurement overnight, when a diurnal signal could not be identified due to the strong tidal signal. In the non-tidal part of the river, we had insufficient night-time data to make a statement about a day-night difference for $pCO_2$ and DO."

*P9, L30 – P10,L1: How did you calculate those gas exchange velocities? I see how your calculations compare well to the A11 model, but R01 model seems to be quite far off. Are those the results for the whole river system?*

The calculation of gas exchange velocities is now described in more detail in Section 2.3. We have changed our approach according to a major comment by Reviewer 2. We now present the three parameterizations as equally valid and present the range of values that they give (average, minimum, maximum).

*P10, L3-5: Those emission rates refer to the entire observed river network? Did you weight the emission rates along the longitudinal profile by stream width?*
We have now calculated emission rates separately for peat, non-peat area and for the delta and calculated emissions (g/month) using the water surface area in each of the categories using stream widths from the GRWL database.

**Comments by Reviewer #2**

*General comment: The manuscript focus on an important topic that I believe is suitable for publication in Biogeosciences. The transport and emission of carbon/GHGs from river networks has repeatedly been concluded during the last decade as a highly significant component when for example estimating landscape C budgets at various scales and biomes. Although the importance is well-recognized, I would claim that relatively little is known about large rivers and their source contribution of atmospheric $CO_2$. The knowledge that exists is largely restricted by the spatiotemporal resolution of the measurements or by using data being based on indirect measurements of $pCO_2$. There is also a clear bias in existing data-sets towards northern hemisphere river networks and with limited information of tropical rivers, especially south-east Asian ones. In this context this study aims to fill an important gap in our understanding concerning large scale drivers of aquatic C in river networks. The influence of peat deposits in the catchment on the pGHG in the water has been shown for various biomes and river network sizes but more extensive investigations are needed. Hence, this is a highly relevant topic especially for a tropical region like this.*
*Although the aim of manuscript is important I have some concerns on how suitable the manuscript is for publication in its current form. My main concerns are: 1) How the actual emissions are calculated. I understand that this is a data scarce region but the way the authors have estimated the emissions is not especially convincing. The author's measure $pCO_2$ in a satisfactory way but the entire k calculation component feels very shaky.*
*No actual measurements of any of the input parameters are conducted. A vague estimate of a fixed water velocity is used in combination with modelled wind data. Three different k parameterizations are then used gaining slightly, to very, different outputs. The model producing intermediate k estimates are then used without any stronger further motivation. The whole procedure feels as I already said very shaky, without knowing anything about the river, investigating seasonal differences in emissions and then using a fixed water velocity sounds for example very strange. On top of these vague calculation steps there are no uncertainty estimate of the calculated emissions (or lateral exports of inorganic and organic C!!). To describe and estimate this in a transparent way would be a requirement in my eyes, especially due to the scarcity in data for the k calculations. If this is problematic to handle, one suggestion is to skip the emission data and solely present the $pCO_2$ patterns and how it varies with wet and dry season and the influence of peatlands. Personally I think this would be the way to go and would be highly interesting in itself. 2) I am not totally*

*convinced of the interpretations of the 13C-DIC data, I am surprised by the generally high 13C-DIC values, the authors claim that the contribution by carbonate containing bedrock to the riverine DIC is minimal in the area and that the river is affected by tidal water sustaining the estuary with marine DIC. That is likely correct but the high 13C-DIC is found even in upstream non-peat area, is the evasion the sole explanation for that? Maybe not relevant, but what about methane production, I understand that methane might have been included in the original plan, but if methane in the peatlands is mainly produced by CO2 reduction this will heavily influence the 13C of the CO2 being delivered to the river (See Campeau et al. 2018 for example). Overall, I find the interpretation of the 13C-DIC data quite short and not as well developed as it could be. 3) Is it really correct to talk about seasonality when just two measurement campaigns are conducted, i.e. wet and dry season? I am not familiar with the region but to call something seasonality or similar would in my mind require a higher sampling resolution in time.*

1) As argued in our Author Comment, we kept the CO2 flux estimates as part of the manuscript, but we improved the discussion of uncertainties and the presentation of the "k-story". We now present the three k-parameterizations as equally valid, and since we point out that the mere choice of a k-parameterization is the largest source of uncertainty (Supplement), we present the range of values that they give (average, minimum, maximum) and discuss it as a range of uncertainty.

2) We have included a Keeling plot in Figure 4, improving our ability to discuss DIC sources. We have also significantly changed the discussion of the isotopic composition of DIC in Section 4.2.1. Factors like methanogenesis and pH are now explicitly mentioned, accompanied by additional references.

3) In Section 4.2.2, we added the following sentence: "Note that since our data was collected during two single surveys, they represent only a snapshot and do not allow strong claims about seasonality."

*Detailed comments:*
*P3 Ln 1-10, there is a mix of wetland and peatland, consistency or a clear separation would be good.*

We rewrote this paragraph: "Borges et al. (2015) established a relationship between wetland extent and $pCO_2$ for African rivers. Wit et al. (2015) presented an analog synthesis for Southeast Asian rivers, which flow through peatlands. Peatlands are a special type of wetland, where organic matter accumulates at rates that make them the most effective terrestrial carbon store on a millennial timescale (Dommain et al., 2011). Southeast Asian peatlands store 68.5 Gt carbon (Page et al., 2011). The highest riverine dissolved organic carbon (DOC) concentrations reported so far were found in Southeast Asian peat-draining rivers (Alkhatib et al. 2007; Moore et al., 2011; Müller et al., 2015), with an annual average of 68 mg $L^{-1}$ DOC found in an undisturbed peat-draining river (Moore et al., 2013). Because of these high DOC concentrations, Indonesian rivers may account for 75 % of the DOC flux into the South China Sea (SCS) while accounting for 39 % of the discharge (Huang et al., 2017). Surprisingly, $CO_2$ emissions from these rivers are not exceptionally high (Müller et al., 2015; Wit et al., 2015). This is attributed to a short residence time of the organic matter in the river, allowing little time for decomposition, and the resistance of peat-derived carbon to bacterial degradation. Nevertheless, the $CO_2$ flux from peat-draining rivers to the atmosphere increases with

increasing peat coverage in the river basin (Wit et al., 2015), showing that these ecosystems exert an important influence on a river's carbon budget."

*P3 Ln 11, Odd formulation and scientifically a bit weird. To claim that something is the highest worldwide is only true until someone else present a higher number. I would recommend to be more open in this formulation.*
We rephrased: "The highest riverine dissolved organic carbon (DOC) concentrations reported so far were found in Southeast Asian peat-draining rivers (Alkhatib et al. 2007; Moore et al., 2011; Müller et al., 2015), with an annual average of 68 mg L-1 DOC found in an undisturbed peat-draining river (Moore et al., 2013)."

*P5 Ln 20-25 and 30, what about correction for salinity on the pCO2 and emissions?*
Please see our author comment for a response to this question. One thing we changed with regard to this comment was adding the sentence: "In-situ $k$ is dependent on in-situ salinity and temperature and was calculated from $k_{600}$, exploiting its relationship with the Schmidt number (Wanninkhof, 1992)." in Section 2.3.

*P6 Ln 8-10 Isn't water velocity dependent on discharge, why is a fixed value used???*
We justified this in our author comment, in the revised manuscript, we present this justification in the Supplement.

*P6 Ln 10, Is there no wind data to validate this modeled data with? How accurate is the wind data compared to conditions over the river is tricky to judge. Feels very vague and uncertain!!!*
Please see our author comment for justification. In the revised manuscript (Section 2.3), we wrote: ". For $u_{10}$, on-site wind speed data was unfortunately not available. In such cases, other authors (e.g., Bouillon et al., 2012; some estuaries in Chen et al., 2013) have resorted to gridded wind data from the NOAA NCEP NCAR Reanalysis product (Kalnay et al., 1996). While we acknowledge the uncertainty introduced by using gridded data instead of in situ wind speed, we used this product as well, as the best one available for our study area."

*P7 Ln 29-30, a bit odd that POC was measured but not DOC. Hard to redo the study but how relevant are the literature DOC values for this study, please motivate better!*
Please see our author comment for a reply. In the revised manuscript (Section 2.5), we now wrote: "For DOC, we used the DOC concentrations reported by Martin et al. (2018). This data was acquired during 2017 downstream of Kanowit. Only freshwater values were considered (average for the wet and dry season: 2.0 mg L$^{-1}$ and 2.1 mg L$^{-1}$)."

*P8 Ln 25, please clarify what pH that is for wet resp. dry season.*
We now wrote: "The Rajang River was slightly acidic (6.7 (wet) and 6.8 (dry), area-weighted mean for the peat and non-peat area, see Table 2)…"

*P9 Ln 10-12, was not the purpose to investigate if the peatlands have an influence on the pCO2 in the river. Feels a bit strange then to say that too few 13C-DIC samples were taken.*
Please find our response in the author comment that we posted. No changes were made to the manuscript with regard to this comment.

*P9 Ln 20, here and elsewhere, what is "distributaries", isn't just tributaries enough???*

Please find our response in the author comment that we posted. No changes were made to the manuscript with regard to this comment.

*P9 Ln 21-23, important sentence but feels more like discussion than result!!*
The second part of the sentence was deleted. Tidal variability is further discussed in the appropriate Section 4.2.3 and in the Supplement.

*P9 Ln 27-28, again, feels more like discussion to me.*
We did not make any changes with regard to this comment. Please see our author comment for justification.

*P10 L4, what does the +-0.52 and +-0.45 mean? Some kind of uncertainty or just spread? Please clarify in the methods. The emission rates (and lateral exports of C) are hard to get a feeling of, how uncertain are they? Impossible to judge for the moment.*
We now clarified in Section 2.5: "Averages of measured parameters are reported ± 1 standard error unless stated otherwise. Errors for calculated parameters (e.g., river loads, see below) were determined with error propagation. For fluxes and derived quantities, we report the mean, minimum and maximum from the three *k*-parameterizations."

*P10 Ln 17-20, Feels from a reader perspective a bit odd to start to say that the findings are the same as found in other studies. I think the authors could "sell" their study better than that. It is important information but I would not place it first in the discussion.*
We rewrote this paragraph, those three lines were deleted as they were redundant after changing the paper.

*Also, maybe a matter of personal taste, but why not start with the main focus of the manuscript in the discussion (pCO2 patterns and maybe emissions if included), the SPM and POC story is secondary as I see it.*
We kept the previous structure, please see our author comment for justification.

*P12 Ln 11-14, Likely true but there is also a strong fractionation in 13C-DIC related to changes/differences in pH which could be up to ca 10 per mille.*
In the course of changing the d13C-DIC discussion, we have included this aspect: "With regard to in-stream processes, photosynthesis increases and respiration decreases $\delta^{13}C$-DIC (Campeau et al., 2017). Due to the high turbidity, it can be assumed that photosynthesis in the Rajang River is negligible. In contrast, the correlation of DO and $pCO_2$ (Fig. 5) suggests that respiration is important. This assumption is supported by the negative correlation of $\delta^{13}C$ and DIC for freshwater samples, because with increasing DIC, $\delta^{13}C$ values get more depleted, suggesting that organic carbon (with a $\delta^{13}C$ of around -26 ‰ for C3 plants, Rózanski et al., 2003) is respired to $CO_2$ within the river. However, we observed overall relatively high $\delta^{13}C$ values. Two processes are likely to be responsible for the downstream increase in $\delta^{13}C$: (1) Methanogenesis and (2) evasion of $CO_2$. (1) Campeau et al. (2018) observed a strong relationship between $CH_4$ concentration and $\delta^{13}C$-DIC in a boreal stream draining a nutrient-poor fen, suggesting that fractionation during methanogenesis leads to an increase in $\delta^{13}C$-DIC. As the peat soils in the Rajang delta are also anaerobic and nutrient-poor, it is likely that methanogenesis plays a role there as well. This is consistent with high reported soil $CH_4$ concentrations of up to 1465 ppm in a peat under an oil palm plantation in Sarawak (Melling et al., 2005). It would therefore be of high interest to investigate $CH_4$ concentrations in the Rajang

River in the future. (2) $CO_2$ evasion is also known to lead to a gradual increase of $\delta^{13}C$-DIC values until in equilibrium with the atmosphere (with $\delta^{13}C$-DIC around +1 ‰, Polsenare and Abril, 2012; Venkiteswaran et al., 2014; Campeau et al., 2018). Due to intracarbonate equilibrium fractionation, dissolved $CO_2$ is more depleted in $\delta^{13}C$ than the other carbonate species. Thus, if it is removed, $\delta^{13}C$ of the remaining DIC increases. This effect depends on pH and is more pronounced in near-neutral waters and less strong in very acidic water (Campeau et al., 2018). We sampled the lower river reaches downstream of Kapit, which corresponds to approximately the last 200 kilometers of the river. In addition, the terrain is much steeper upstream of Kapit than in the lower river reaches, so that $CO_2$ fluxes to the atmosphere are presumably much higher due to higher turbulence. This means that a large fraction of $CO_2$ had likely already been emitted from the river surface before reaching Kapit, leading to the observed high $\delta^{13}C$-DIC values."

*Table 2.What is the +- of the emissions, the SE of the mean? I.e. some kind of measure of the spatial variability? Is this driven by something else than just variability in pCO2? Is k fixed for all data? According to the methods I get this feeling. Please clarify in the methods.*
We have now clarified in the Methods section 2.5; in addition, the information "+- SE" is found in the Table caption.

[revised manuscript text omitted]

**1 Pressure correction**

**1.1 Failure of the Internal Pressure Sensor**

The Li-820 maintains a stable cell temperature and corrects the absorptance of $CO_2$ based on a measurement of the pressure in the cell. During the cruise in August 2016, a failure of the internal pressure sensor occurred on August 25, 2016 at 22:53 GMT. The failure was evident, because the cell pressure reading dropped from a relatively stable value of 102 kPa to 62.57 kPa within 10 seconds. Even when all tubes and pumps were removed and the Li-820 cell pressure was allowed to adjust to ambient pressure, the reading did not change. The internal pressure correction that the Li-820 performs was thus based on the false reading of a cell pressure of 62.57 kPa. The setup had not been changed, and the cell pressure before the failure had been at a stable level of approximately 102.2 kPa. Consequently, the pressure correction done by the Li-820 was reversed and performed again assuming an internal cell pressure of 102.2 kPa for the time after the failure of the pressure sensor.

**1.2 Pressure Correction in the Li-820**

The $CO_2$ mole fraction in the Li-820 is computed from a pressure-corrected measurement of absorptance. The pressure correction is performed by multiplication of the absorptance $\alpha_c$ with an empirically determined correction function (Li-820 Manual, Eq. 4-4):

$$\alpha_{pc} = \alpha_c g_c(\alpha_c, P) \tag{1}$$

$$g_c(\alpha_c, P) = \begin{cases} X \, for \, P < P_0 \\ 1 \, for \, P = P_0 \\ \frac{1}{X} \, for \, P > P_0 \end{cases} \tag{2}$$

with $P_0 = 99 kPa$ and $X = \dfrac{1}{\frac{1}{b_1(p-1)} + \frac{1}{\frac{1}{b_2+b_3 p}+b_4}} + 1$ (Li-820 manual, Eq. 4-6). In

this equation, $p = \frac{P_0}{P}$ or $\frac{P}{P_0}$, with $p > 1$.

So

$$\alpha_{pc} = \begin{cases} \alpha_c \cdot X \, for \, P < P_0 \\ \frac{\alpha_c}{X} \, for \, P > P_0 \\ \alpha_c \, for \, P = P_0 \end{cases} \tag{3}$$

**1.3 Correction for false cell pressure**

In order to correct for the false cell pressure $P_{meas}$, the absorptance $\alpha_c$ has to be computed. Then, the pressure-corrected absorptance $\alpha_{pc}$ has to be calculated using the corrected pressure $P_c$. $P_c$ was taken to be the average cell pressure during the measurements before the pressure sensor failed, which was 102.18

kPa. This value was considered a good approximation, as the cell pressure in the Li-820 was fairly stable during the time of measurements while the pressure sensor was still functioning.

The inverse function which allows calculation of the pressure-corrected absorptance from the mole fraction is given as

$$\alpha_{pc} = \frac{a_1 C}{a_2 + C} + \frac{a_3 C}{a_4 + C} \tag{4}$$

with $a_1 = 0.3989974$, $a_2 = 5897.2804$, $a_3 = 0.097101982$, $a_4 = 596.49981$ (Li-820 Manual, Eq. 4-7).

In order to calculate the absorptance $\alpha_c$ from $\alpha_{pc}$, Equation 3 has to be rearranged and solved for $\alpha_c$. The solutions are:

$$\alpha_c = \begin{cases} -\frac{P_1}{2} - \sqrt{(\frac{P_1}{2})^2 - Q_1} \, for \, P < P_0 \\ -\frac{P_2}{2} - \sqrt{(\frac{P_2}{2})^2 - Q_2} \, for \, P > P_0 \\ \alpha_c \end{cases} \tag{5}$$

whereas
$P_1 = \frac{\alpha_{pc} m b_5 (n+b_4) - \alpha_{pc} + b_5^2 (n+b_4)(m+1)}{-b_5 (n+b_4)(m+1) + 1}$
$Q_1 = -\frac{\alpha_{pc} m b_5^2 (n+b4)}{-b_5 (n+b_4)(m+1) + 1}$
$P_2 = \frac{b_5^2 m (n+b_4) - \alpha_{pc} + \alpha_{pc} b_5 (1+m)(n+b_4)}{1 - b_5 m (n+b_4)}$
$Q_2 = -\frac{\alpha_{pc} b_5^2 (1+m)(n+b_4)}{1 - b_5 m (n+b4)}$
with $m = \frac{1}{b_1 (p-1)}$ and $n = \frac{1}{b_2 + b_3 p}$.

From $\alpha_c$ and $P_c$, the corrected $\alpha_{pc,c}$ is calculated according to Equation 1. $C_c$ is calculated according to the manual:

$$C_c = \frac{D - (a_2 + a_4)\alpha_{pc,c} - \sqrt{A^2 \alpha_{pc,c}^2 + B\alpha_{pc,c} + D^2}}{2(\alpha_{pc,c} - a_1 - a_3)} \tag{6}$$

whereas $A = a_2 - a_4$, $B = 2A(a_1 a_4 - a_2 a_3)$ and $D = a_3 a_2 + a_1 a_4$ (Li-820 Manual, Eq. 4-10 and 4-11).

**2 Salinity Interpolation**

Salinity was only available at the stations (15 in the wet season, 34 in the dry season). However, in order to be able to interpret $O_2$ and $CO_2$ data, it is useful to know their distribution along a salinity gradient. Therefore, salinity in the estuary was spatially interpolated. Since the saltwater intrusion limit was presumably different between wet and dry season, interpolation was performed for the entire area under tidal influence (downstream of Kanowit). Beyond that point, salinity was measured to be zero.

Since some points for which interpolation was desired lay outside the area covered by our measurements, we added three reference points to better constrain the grid to be interpolated. The coordinates of these points were:

$$(2.0, 2.5, 3.5), (111.0, 111.0, 111.8) \tag{7}$$

These reference points all lie within the South China Sea off the coast of Sarawak. The salinity value ascribed to them was 33 according to our own measurements and those of Wang et al. (2014) for the Southern South China Sea. Interpolation was achieved with the Scipy Interpolation package for Python (scipy.interpolate.griddata) using linear interpolation. Figure 1a shows the points used for interpolation, Figures 1b and c show the results for the wet and dry season, respectively.

[Figure]

Figure 1: Data points used for interpolation (a), results for the wet (b) and dry (c) season. Interpolated salinity is shown in graduated colors, actual measurements are shown as squares.

**3 Water surface area in the delta**

We used stream widths for the Rajang River from the GRWL (Global River Widths from Landsat) Database (Allen and Pavelsky, 2018). The length of the river segments was determined using ArcMap 10.5 and multiplied by the mean river width. Missing parts were manually delineated using a georeferenced Landsat satellite image (Fig. 2, source of the Landsat image: `https://landsat.visibleearth.nasa.gov/view.php?id=91787` (last access: Oct 9th, 2018)). The total water surface area in the Rajang catchment was calculated at 755 km$^2$ or 1.5% of the catchment area.

[Figure]

Figure 2: River segments used to determine the water surface area of the Rajang River. The close-up shows manually delineated segments in the delta using a georeferenced satellite image.

**4    Tides**

Tidal variability was only observed at the river mouth. The Figure shows water level from close-by stations and the measured $pCO_2$. The rectangle marks the only stationary measurement, which was performed in Sarikei overnight and covers one tidal cycle. For all other data, spatial and temporal variations are overlapping, because the ship was moving. Tidal variability in $pCO_2$ cannot be observed at all upstream of Sibu or in the Igan distributary. In the Paloh and Rajang distributary, variability in $pCO_2$ is high, but this is partly attributed to mixing with sea water.

[Figure]

Figure 3: Measured pCO₂ in January 2016 and water level in different river reaches.

[Figure]

Figure 4: Measured $pCO_2$ in August 2016 and water level in different river reaches.

**5  Gas transfer velocity considerations**
[Figure]

The choice of a $k$-model has a big impact on the calculated $CO_2$ fluxes. Therefore, three different models are compared (Table 1). Additional uncertainty arises from the input data. Those models depend on wind speed and water flow velocity. In the absence of in-situ wind speed data, we had to use the NOAA NCEP Reanalysis product; however, wind speed on-site might differ from these values, which would impact our results for $k$. Secondly, we used two literature values for the Rajang River's water flow velocity $w$, that is, one fixed value for both seasons. According to Raymond et al. (2012), $w$ scales with discharge $Q^{0.29\pm0.01}$. During the peak of the monsoon season in January, $Q$ is approximately 50 % higher than average discharge, which would mean that $w$ would be enhanced by 12 %. If we consider this the variability in $w$ ($w = 0.9 \pm 0.1$ m s$^{-1}$), it would add an uncertainty of 4 % to $k_{600,B04}$. However, the deviation among the different $k$-models is much larger than that (Table 1), so the biggest source of uncertainty isn't the input data, but the choice of a $k$-model. Table 1 presents a comparison of three different k-parameterizations.

| | | $k_{600}$ | | | $FCO_2$ | | |
| --- | --- | --- | --- | --- | --- | --- | --- |
| | | B04 | A11 | R01 | B04 | A11 | R01 |
| non-peat | wet | 8.23 | 8.51 | 2.32 | $1.9 \pm 0.2$ | $2.0 \pm 0.2$ | $0.5 \pm 0.0$ |
| | dry | 9.57 | 12.19 | 2.79 | $2.0 \pm 0.3$ | $2.6 \pm 0.4$ | $0.6 \pm 0.1$ |
| peat | wet | 8.23 | 8.51 | 2.32 | $2.3 \pm 0.2$ | $2.4 \pm 0.2$ | $0.7 \pm 0.1$ |
| | dry | 9.57 | 12.19 | 2.79 | $2.7 \pm 0.1$ | $3.5 \pm 0.2$ | $0.8 \pm 0.0$ |
| delta | wet | 8.23 | 8.51 | 2.32 | $2.3 \pm 1.0$ | $2.4 \pm 1.0$ | $0.7 \pm 0.3$ |
| | dry | 9.57 | 12.19 | 2.79 | $2.3 \pm 1.4$ | $3.0 \pm 1.7$ | $0.7 \pm 0.4$ |

Table 1: Comparison of the results for different k-parameterizations. B04: Borges et al. (2004), A11: Alin et al. (2011), R01: Raymond & Cole (2001). $\pm$ represents the spread of the data (derived from the spread of the $pCO_2$). In the main manuscript, the average of the three values is used, minimum and maximum are reported alongside.

**6  Mixing model**

We used a simple mixing model to estimate the theoretically possible contribution of the peatlands to river $pCO_2$. The model consists of two subsequent steps. First, the mixing of two water bodies was simulated, one with a $pCO_2$ of 2434 $\mu$atm and a pH of 6.8 (Rajang River), and the other with a $pCO_2$ of 8100 $\mu$atm and a pH of 3.8 (representing peat-draining rivers according to Müller et al., 2015). The DIC and TA of these water bodies were calculated using $CO_2$Sys. DIC and TA of the mixture were calculated as

$$DIC_{S=0} = (1 - pc) \cdot DIC_1 + pc \cdot DIC_2 \tag{8}$$

and

$$TA_{S=0} = (1 - pc) \cdot TA_1 + pc \cdot TA_2, \tag{9}$$

whereas pc is the peat coverage in the basin as the river flows downstream and passes through more and more peat areas (pc=0...0.11). As a next step, for pc>0.03, mixing with saltwater was taken into account. It was assumed that pc=0.03 correpsonds to S=0 and that pc=0.11 corresponds to S=32 and within this range, salinity increased linearly with increasing peat coverage. This is obviously a simplification, but since the model has only illustrative purposes, it seemed sufficient. DIC and TA were then calculated with a normal end-member mixing model:

$$DIC = \frac{DIC_{S=32} - DIC_{S=0}}{32} \cdot S + DIC_{S=0} \tag{10}$$

and

$$TA = \frac{TA_{S=32} - TA_{S=0}}{32} \cdot S + TA_{S=0}, \tag{11}$$

whereas $TA_{S=32} = 2324 \mu mol L^{-1}$ and $DIC_{S=32} = 2347 \mu mol L^{-1}$ according to our measurements. $pCO_2$ was calculated from TA and DIC using $CO_2Sys$. The mixing model and the results are shown in Figure 4.

[Figure]

Figure 5: Mixing model flow chart and plot of the results for theoretically possible $pCO_2$ if peat is the only source of $CO_2$ in the delta.

**7    Supplementary Data**

The data used in this manuscript are available as a separate excel workbook.

---

## Author Response (AR2)

*P3, L19, remove "However", is not needed.*

Deleted.

*P10, L10-15, decrease the number of digits on the k estimates. Because these are really estimates, which the authors also discuss the uncertainty of. So one decimal is enough.*

Done.

*P11, L21, Should both rivers on this row be named Rajang? Is the Huang et al study from the same river? In such case is should be highlighted and expressed differently.*

Yes, Huang's data is from the Rajang. We changed the wording:

"DIC concentrations measured during our dry season survey were comparable to those determined by Huang et al. (2017) for the Rajang River (201 µmol L$^{-1}$ and 487 µmol L$^{-1}$). Their values are based on 7 measurements taken between 2005 and 2009 downstream of Sibu (pers. comm.)."

*P11, L29, remove tracking on comma sign*

Done

*P12, L15, add starting parenthesis for the Rozanski ref*

The parentheses start in line 14: (with a $\delta^{13}$C of around -26 ‰ for C3 plants, Rózanski et al., 2003)

*P12, L24, is really equilibrium with the atmosphere at +1 per mille?*

Atmospheric CO2 has a d13C of approx. -8‰. Equilibrium fractionation between CO2 and HCO3 is around 9‰, so HCO3 from atmospheric CO2 should be slightly positive. Thus, in near-neutral water, where the DIC pool is dominated by HCO3, d13C-DIC of DIC in equilibrium with the atmosphere should indeed be slightly positive. +1‰ was the value yielded by the model of Polsenare & Abril 2012. Of course, this entails certain assumptions, and the actual value is obviously dependent on pH, temperature and the atmospheric d13C-CO2, so we changed the wording to "slightly positive" in order to not let the value of +1‰ appear as commonly accepted atmospheric equilibrium d13C-DIC.

*P13, L16, yes, but pCO2 is partly at least controlling pH. Keep track on the "hen and the egg".*

We rephrased this paragraph – we do not claim that pH drives pCO2, but that one cannot be understood without the other. The paragraph now reads:

"A meaningful comparison is also the one between the Rajang River and the Indragiri River, Indonesia, because they have a similar peat coverage (Rajang: 11%, Indragiri: 12%) and peat coverage has

previously been considered as a good predictor of river $CO_2$ emissions (Wit et al., 2015). However, $pCO_2$ in the Indragiri (5777 µatm) was significantly higher than in the Rajang, which was associated with a lower pH (6.3, numbers from Wit et al., 2015). A simple exercise using $CO_2$Sys illustrates how important it is to consider both pH and $pCO_2$ when comparing different peat-draining rivers. At the given temperature, salinity and pH, the $pCO_2$ of 5777 µatm in the Indragiri corresponds to a DIC value of 327 µmol $L^{-1}$. At a hypothetical pH of 6.8, as measured in the Rajang River, this DIC value corresponds, under otherwise unchanged conditions, to a $pCO_2$ of 2814 µatm – which is very close to the average values measured in the peat area of the Rajang River. The close coupling of pH and $pCO_2$ implies that peat coverage in a river basin is insufficient as sole predictor of $CO_2$ fluxes. Rather, the relationship between peat coverage and $pCO_2$ must be viewed in the context of the rivers' pH, and drivers of pH must be more carefully considered. Note also that peat coverage is usually reported for the entire catchment (e.g., Wit et al, 2015; Rixen et al., 2016) and does not reveal how much peat is found in estuarine vs. freshwater reaches, which complicates comparisons further."

*P13, L17, the difference in 13C-DIC is also a function of the different pH, so not solely a function of different C sources, although maybe related.*

We rephrased:

"$\delta^{13}$C-DIC in the Indragiri was lower (-16.8 ‰, Wit, 2017) than in the Rajang (-7.0 ‰). While this can be partially explained by the different pH, it can also be interpreted as a greater contribution of respiratory $CO_2$ in the Indragiri, while the Rajang might be more strongly influenced by weathering. This would also help to explain the higher pH in the Rajang River."

*P15, L16, yes, but you have now also added the new 13C-DIC interpretation paragraph on P12 where you discuss the methanogenisis and its potential source. I find this statement a bit redundant in that context.*

The d13C-DIC values of the peat-draining rivers is a lot lower than in the Rajang river (-29‰ vs. -7‰), so if there was a significant contribution of DIC from peat-draining rivers, we would expect to see an effect despite the other processes governing d13C-DIC. However, this was not observed. We changed the wording of this paragraph and clearly separated the last part of this paragraph, to make it clear that it is a conclusion derived from a), b) and c) and not only from c):

"If the peatlands acted as a significant source of $CO_2$ to the Rajang River, it would be expected that this had an impact on the $\delta^{13}$C-DIC values. In the peat-draining Maludam River, $\delta^{13}$C-DIC averaged -28.55 ‰ (Müller et al., 2015). Thus, the influx of peat-draining tributaries to the Rajang River (with an average $\delta^{13}$C-DIC of -7‰) would theoretically decrease $\delta^{13}$C-DIC. This was not observed. Instead, the main source identified by the Keeling plot was consistent with groundwater input with a $\delta^{13}$C-DIC of -18‰. However, given the constraints on the applicability of the Keeling plot method discussed above, the effect of DIC inputs from peat on the $\delta^{13}$C-DIC might simply not be detectable. Other processes which influence $\delta^{13}$C-DIC as discussed above might also prevent identification of a peat signal in the $\delta^{13}$C-DIC data.

In summary, we were unable to discern a large impact of peatlands on the DIC budget of the Rajang River. It is possible that, because the peatlands are located close to the coast in this system, mixing with sea water occurs before significant effects on the $pCO_2$ are theoretically possible. This means that not only the peat coverage in the catchment is relevant, but also how much of this peat is found in estuarine

reaches. These findings support the arguments of Müller et al. (2015) and Wit et al. (2015) that material derived from coastal peatlands is swiftly transported to the ocean, explaining why peat-draining rivers may not necessarily be strong sources of $CO_2$ to the atmosphere."

2) Editor's comments

*Dear Denise and co-authors,*
*Thank you for the revised version of your manuscript and the detailed author replies. As you can see, the revised ms has been re-evaluated by one of the original referees, who feels that you have adequately addressed their suggestions and concerns. I generally agree with this, but do have a few concerns on the Keeling plot approach now added to the interpretation of the d13C-DIC data. This is an elegant approach but all too often used for settings in which it is not applicable – keep in mind that the approach essentially stems on a mass balance for a reservoir to which a certain quantity is added. Since we are looking here at a concentration gradient across a salinity gradient, the slopes of a Keeling plot in this case offer not new information on the sources of the DIC. Such mixing plots can be useful to find a d13C value when there is a net input of C to an existing C pool in a given system with clear boundaries. Hence, plotting the data collected along the salinity gradient is not informative – it basically returns the weighted average d13C value of all data points. You currently only use the approach in quantitative terms for the freshwater data, which is less problematic but it nevertheless still assumes that the data can be interpreted as representing data from a gradual addition of DIC to an initial DIC pool – which would require some discussion to convince the reader that this is applicable.*
*Furthermore, on p14, line 19 and further, you mention "the y-intercept of the Keeling plot for freshwater samples suggests that the initial freswhater source has a d13C-DIC of -18.6 ‰" – this is not really consistent with how a Keeling plot should be interpreted; the intercept represents the d13C value of the DIC added to the systems' initial DIC pool.*
*I would recommend to reflect on the applicability of the Keeling plot approach to your data – and rework this section of the manuscript.*

We agree that the basic assumptions of the Keeling plot method must be borne in mind when this method is applied. In the revised manuscript, we add our main assumptions and considerations regarding this method in the Methods section (Section 2.5). We now wrote:

"A Keeling plot (Keeling, 1958) was used to explore possible sources of DIC in the Rajang River. The Keeling plot method has been used to determine the isotopic signature in $CO_2$ from ecosystem respiration (Pataki et al., 2003; van Asperen et al., 2017) by plotting $\delta^{13}C$ of $CO_2$ in an air sample versus the inverse $CO_2$ concentration. From the y-intercept of a linear regression, the isotopic signature of the source can be determined (Keeling 1958; Pataki et al., 2003). The Keeling plot method assumes mixing of only two components: One background component (e.g., atmospheric background) and one additional source (e.g., respiration). While originally designed for atmospheric research, it has also been used in studies exploring possible sources of DIC in stream water (Campeau et al., 2017; 2018). However, it has to be interpreted with caution, because rivers are open systems where the basic assumption of two-component-mixing is easily violated. When we interpret the Keeling plot in this study, we look at mixing between river DIC and added DIC from groundwater and peat-draining rivers. We thus have to assume that DIC in the main river does not change over the time of our measurements. Since the water is continuously replenished from upstream, this assumption is valid as long as the

upstream (background) source's isotopic signature remains the same. This can be assumed during a relatively short sampling period (7 days). Nevertheless, we can only consider freshwater data, as mixing with sea water already constitutes a third component. Therefore, we show only freshwater data in our Keeling plot and caution that its interpretation is based on the relatively strong assumption that we can view the river's main stem $\delta^{13}$C-DIC as constant over the sampling period."

In addition, we removed the estuarine data from Figure 4 and in Section 4.2.1, we rephrased:

"Multiple sources and processes are likely to influence $\delta^{13}$C-DIC in the Rajang River. To start with, the y-intercept of the Keeling plot for freshwater samples suggests that the DIC added to the main river has a $\delta^{13}$C of -18.6 ‰, which is consistent with $\delta^{13}$C values of bicarbonate from silicate weathering with soil $CO_2$ from C3 plants (-22.1 to -16.1 ‰, Das et al., 2005), i.e. groundwater input."

*One further suggestion:*
*-p16, L8-17: this is a somewhat circular line of reasoning: pH and pCO2 are intrinscically linked - see comment by Referee #2 ("the hen and the egg").*

Please see our response to the Referee's comment above – we reworked this paragraph. Our only intention was to make a point that pCO2 cannot be interpreted without the knowledge of pH. We hope that this is now more clearly expressed in the manuscript.

[revised manuscript text omitted]